# *Campomanesia adamantium* O Berg. fruit, native to Brazil, can protect against oxidative stress and promote longevity

Laura Costa Alves de Araújo[1], Natasha Rios Leite[1], Paola dos Santos da Rocha[1], Debora da Silva Baldivia[1], Danielle Araujo Agarrayua[2], Daiana Silva Ávila[2], Denise Brentan da Silva[3], Carlos Alexandre Carollo[3], Jaqueline Ferreira Campos[1], Kely de Picoli Souza[1], Edson Lucas dos Santos [1]*

1 Research Group on Biotechnology and Bioprospecting Applied to Metabolism (GEBBAM), Federal University of Grande Dourados, Dourados, Mato Grosso do Sul, Brazil, 2 Research Group in Biochemistry and Toxicology in *Caenorhabditis elegans*, Federal University of Pampa, Uruguaiana, Rio Grande do Sul, Brazil, 3 Laboratory of Natural Products and Mass Spectrometry, Federal University of Mato Grosso do Sul, Campo Grande, MS, Brazil

* edsonsantosphd@gmail.com

**Data Availability Statement:** All relevant data are within the paper and its Supporting Information files.

## Abstract

*Campomanesia adamantium* O. Berg. is a fruit tree species native to the Brazilian Cerrado biome whose fruits are consumed raw by the population. The present study determined the chemical composition of the *C. adamantium* fruit pulp (FPCA) and investigated its *in vitro* antioxidant potential and its biological effects in a *Caenorhabditis elegans* model. The chemical profile obtained by LC-DAD-MS identified 27 compounds, including phenolic compounds, flavonoids, and organic carboxylic acids, in addition to antioxidant lipophilic pigments and ascorbic acid. The *in vitro* antioxidant activity was analysed by the radical scavenging method. *In vivo*, FPCA showed no acute reproductive or locomotor toxicity. It promoted protection against thermal and oxidative stress and increased the lifespan of *C. elegans*. It also upregulated the antioxidant enzymes superoxide dismutase and glutathione S-transferase and activated the transcription factor DAF-16. These results provide unprecedented *in vitro* and *in vivo* evidence for the potential functional use of FPCA in the prevention of oxidative stress and promotion of longevity.

## Introduction

The Cerrado is a global biodiversity hotspot, being recognized as the richest tropical savanna in the world and housing approximately 12,000 species of native plants that have been catalogued, several of which have a strong cultural and economic impact on local communities [1]. Timber, dyeing, ornamental, medicinal, and food species stand out for their regional relevance. Food plant genera have different species that produce edible fruits, with varied shapes, attractive colours, and characteristic flavours [2, 3].

The fruit species of the Cerrado have many and diverse bioactive compounds, and these compounds can be beneficial to human health, representing a potential source of food with

**Funding:** The author(s) received no specific funding for this work.

**Competing interests:** The authors have declared that no competing interests exist.

functional properties to be incorporated into the diet or to be used in the cosmetic and pharmaceutical industries [3, 4]. The fruits are considered excellent sources of natural antioxidant compounds that are important constituents of the human diet. Those compounds are a heterogeneous group of molecules that can donate hydrogen atoms or electrons, and their stable intermediate radicals prevent the oxidation of molecules in the body [5]. The benefits of fruit consumption can be attributed to the presence of specific compounds, such as minerals, fibres, vitamins, phenolic compounds, and flavonoids. All these nutrients are closely correlated with a reduced risk of cardiovascular and chronic diseases [6–8]. The biological activities of a given food are associated with synergistic or antagonistic biochemical interactions between nutrients, promoting physiological responses capable of modulating metabolism in oxidative stress processes [9]. Thus, foods that act in signalling pathways capable of minimizing oxidative stress can modulate and delay the progression of ageing [10–12].

Among these native fruits is *Campomanesia adamantium* O. Berg (Myrtaceae), a fruit tree species found in various regions, especially the Cerrado. It is popularly known as *guavira* or *gabiroba*. The fruits produced by this species are available for a short time during the year, which hinders their production and commercialization.

In folk medicine, the leaves and fruits of *C. adamantium* are used as antirheumatic, antidiarrhoeal, hypocholesterolaemic and anti-inflammatory agents [13]. Scientifically, different parts of this plant have already been described because they have different pharmacological properties. The leaves and roots have anti-leukaemic activity by activating intracellular calcium and caspase-3 and inducing apoptosis [14]. In addition, the roots have antioxidant activities *in vitro* and *in vivo* and cholesterol- and triglyceride-lowering effects [15]. The essential oil of the fruits shows anti-inflammatory and antinociceptive activities [13]. Fruit peels have antihyperalgesic, antidepressant, and anti-inflammatory effects [16] and are also able to inhibit cyclooxygenases 1 and 2 and platelet aggregation [17]. Its fruit pulp is described as having antiproliferative action against murine melanoma cells [18] and *in vitro* antioxidant activity that protects against oxidative stress–inducing agents in a cellular hepatoxicity model [19].

Despite these scientific studies that demonstrate the functional properties of different parts of *C. adamantium*, there are still few studies on the biological and nutraceutical properties of its fruits, the plant part directly consumed by the population. Thus, the objectives of this study were to determine the chemical composition, characterize antioxidant compounds, and evaluate the *in vitro* and *in vivo* antioxidant activity of the *C. adamantium* fruit pulp (FPCA); and to investigate its toxicological parameters and its effects on lifespan in *Caenorhabditis elegans*.

## Material and methods

### Materials

The chemicals were purchased from Sigma-Aldrich: formic acid, 2,2-diphenyl-1-picrylhydrazyl, 2,2'-azinobis-(3-ethylbenzothiazoline-6-sulfonic acid, Juglone (5-hydroxy-1,4-naphthoquinone) 2,6-dichlorophenolindophenol-sodium (DCIP), potassium persulfate, butylated hydroxytoluene (BHT), quercetin, oxalic acid and sodium hypochlorite; Dinâmica: methanol, acetone, hexane, Folin-Ciocalteu, sodium carbonate, aluminum chloride hexahydrate, ascorbic acid and sodium hydroxide; Diversey: Sumaveg®.

### Collection and preparation of *C. adamantium* fruit pulp

The fruits of the species *C. adamantium* were collected in fragments of the Cerrado Biome, located in the municipality of Dourados (S 21° 59' 41.8" and W 55° 19' 24.9"), Mato Grosso do Sul state, Brazil. To obtain the pulp of the *C. adamantium* fruit (FPCA), the fruits were washed in running water to remove impurities, sanitized by immersion in Sumaveg® solution (3.3 g/L

of water) for 15 minutes, rinsed with drinking water, depulped, followed by lyophilization and storage at -80˚C. For the experimental assays, 0.005 g of FPCA was resuspended in 5 mL of sterile ultrapure water and homogenized by constant agitation for 5 minutes. Then, it was placed in light-protected tubes and refrigerated at 4˚C for 24 hours, aiming to achieve better dissolution of the pulp and its chemical constituents. Only after this period, FPCA was used in the experimental analyses, as shown the following flow chart.

## Identification of the constituents by LC-DAD-MS

The sample of FPCA (40 mg) was extracted with methanol and deionized water added 0.1% formic acid (7:3, v/v) (3 mL) for 15 min in the ultrasonic bath. Subsequently, the sample was centrifuged, and the supernatant was filtered on Millex® (PTFE membrane, 0.22 μm) to be injected into the chromatographic system (injection volume 5 μL). The sample was injected on a UFLC Prominence Shimadzu coupled to a diode array detector (DAD) and a mass spectrometer (MicrOTOF-Q III, Bruker Daltonics, Billerica, MA, USA). Kinetex C18 column (2.6 μm, $150 \times 2.1$ mm, Phenomenex) was used for analyses, applying a flow rate of 0.3 mL/min and oven temperature of 50˚C. The mobile phase was composed of deionized water (solvent A) and acetonitrile (solvent B), both added 0.1% formic acid (v/v), and the following gradient elution profile was applied: 0–2 min 3% B, 2–25 min 3–25% B, 25–40 min 25–80% B and 40–43 min at 80% B. For the MS analyses, nitrogen was used as nebulizer gas at 4 Bar, dry gas at 9 L/min, and collision gas. The analyses were acquired in negative and positive ion modes.

## Determination of total phenolic compounds and flavonoids

To determine the levels of phenolic compounds and flavonoids, the FPCA was centrifuged at 5000 rpm for 10 minutes, and the supernatant was used for the analyses.

**Phenolic compounds.** The levels of phenolic compounds present in the FPCA were determined using the Folin-Ciocalteu colorimetric method. For this, 2.5 mL of Folin-Ciocalteu reagent (1:10 v/v, diluted in distilled water) was added to 0.5 mL of FPCA (at a concentration of 500 μg/mL). This solution was incubated in the dark for 5 minutes. Subsequently, 2.0 mL of 14% aqueous sodium carbonate ($Na_2CO_3$) was added and incubated at room temperature for 120 minutes, protected from light. The absorbance was measured at 760 nm using a T70 UV/Vis spectrophotometer (PG Instruments Limited, Leicestershire, UK). A calibration curve with gallic acid (0.0004–0.0217 mg/mL) was used as a standard. The phenolic compounds in the FPCA were expressed as mg gallic acid equivalent (GAE) per gram of pulp. Three independent assays were performed in triplicates.

**Total flavonoids.** To determine the levels of flavonoids in the FPCA, a 2% ethanolic solution of aluminum chloride hexahydrate ($AlCl_3 \cdot 6H_2O$) (4.5 mL) was added to 0.5 mL of pulp (at a concentration of 500 μg/mL), and this solution was kept in the dark for 30 minutes at room temperature. Subsequently, the absorbances were measured at 415 nm (T70 UV/Vis spectrophotometer, PG Instruments Limited, Leicestershire, UK). The calibration curve was prepared using the standard compound quercetin (0.0004–0.0217 mg/mL). The total content of flavonoids in the FPCA was expressed as mg quercetin equivalent (QE) per gram of pulp. Three independent assays were performed in triplicates.

## Determination of lipophilic compounds

For the determination of lipophilic antioxidant compounds β-carotene, lycopene, and chlorophyll a and b, 150 mg of FPCA was vigorously agitated in 10 mL of an acetone-hexane mixture (4:6, v/v) for 1 minute, and then filtered using qualitative filter paper Whatman® Grade 4. The absorbances of the filtrate were measured at 453, 505, 645, and 663 nm. The contents of β-

carotene, lycopene, and chlorophyll a and b were calculated using mathematical equations:

$$\beta - \text{carotene} = 0.216 \times \text{Abs663} - 1.220 \times \text{Abs645} - 0.304 \times \text{Abs505} + 0.452 \times \text{Abs453} \tag{1}$$

$$\text{Lycopene} = -0.0458 \times \text{Abs663} + 0.204 \times \text{Abs645} + 0.304 \times \text{Abs505} - 0.0452 \times \text{Abs453} \tag{2}$$

$$\text{Chlorophylla a} = 0.999 \times \text{Abs663} - 0.0989 \times \text{Abs645} \tag{3}$$

$$\text{Chlorophylla b} = -0.328 \times \text{Abs663} + 1.77 \times \text{Abs645} \tag{4}$$

The results were expressed in mg/100 g of FPCA. Three independent assays were performed in triplicates.

## Determination of ascorbic acid

To determine the concentration of ascorbic acid, 0.5 g of FPCA was vigorously homogenized in 50 mL of oxalic acid. Then, 20 mL of this solution was transferred to a 50 mL volumetric flask and the volume was completed with oxalic acid. The mixture was filtered using qualitative filter paper, Whatman® Grade 4. The filtrate was used to titrate a solution of the indicator (DCIP), 2,6-dichlorophenolindophenol-sodium. The titration was completed when a persistent pink color appeared for 15 s. Ascorbic acid was used as a standard control. The results were calculated based on the following equation and expressed in mg of ascorbic acid/100 g of FPCA:

$$\frac{mg \text{ Ascorbic acid}}{100g_{FPCA}} = \frac{DCIP_{FPCA}}{DCIP_{standard}} \times \frac{100}{M_{FPCA}} \times \frac{(M_{solvent} + M_{FPCA})}{M_{FPCA}} \times \frac{50 \ mL}{10 \ mL} \times F$$

$$F = \frac{M_{AA}}{50} \times \frac{1}{25} \times 10 \tag{5}$$

Where, $DCIP_{FPCA}$ and $DCIP_{standard}$ are the volumes used for titration of the sample and standard, respectively, in mL. $M_{solvent}$ and $M_{FPCA}$ are the respective masses of the solvent and sample, added for sample titration, and an aliquot of the sample in grams. F is the amount of ascorbic acid required to reduce DCIP (mg), and $M_{AA}$ is the mass of ascorbic acid (mg). Three independent experiments were performed in triplicates.

## Antioxidant activity *In vitro*

**DPPH• free radical scavenging activity.** To evaluate the DPPH• (2,2-diphenyl-1-picryl-hydrazyl) free radical scavenging activity, 0.2 mL of the FPCA (0.1–1000 μg/mL) was mixed with 1.8 mL of a DPPH solution (0.11 mM) diluted in 70% ethanol. The mixture was homogenized and incubated at room temperature for 30 minutes, protected from light. The absorbance was measured at 517 nm. Ascorbic acid and butylated hydroxytoluene (BHT) (0.1–1000 μg/mL) were used as reference antioxidants (positive controls). Three independent assays were performed in triplicates. The inhibition curve was prepared, and the IC50 values (concentration required to inhibit 50% of the free radicals) were calculated. The percentage of DPPH• free radical elimination was calculated from the control (0.11 mM DPPH solution) using the following equation:

$$\text{DPPH}^{\bullet} \text{ free radical scavenging activity } (\%) = 1 - \frac{Abs \ sample}{Abs \ control} \times 100 \tag{6}$$

**ABTS•+ radical decolorization assay.** The ABTS•+ (2,2'-azinobis-(3-ethylbenzothiazo-line-6-sulfonic acid) radical scavenging capacity was performed by mixing 5 mL of the ABTS solution (7 mM) with 88 μL of potassium persulfate solution (140 mM). The mixture was kept at room temperature, protected from light, for 12–16 hours. Then, the solution was diluted in absolute ethanol to obtain an absorbance of $0.70 \pm 0.05$ at 734 nm. Subsequently, 20 μL of the FPCA (0.1–1500 μg/mL) was mixed with 1980 μL of the ABTS•+ radical solution. The solution was homogenized and incubated for 6 minutes at room temperature, protected from light. The absorbance was measured at 734 nm. Ascorbic acid and BHT were used as reference antioxidants (positive controls). Two independent assays were performed in triplicates. The inhibition curve was prepared, and the IC50 values were calculated. The percentage of ABTS•+ inhibition was determined according to the following equation:

$$\text{ABTS}^{•+} \text{ radical inhibition (\%)} = \left( \frac{Abs\ control - Abs\ sample}{Abs\ control} \right) \times 100 \qquad (7)$$

## *In vivo* assays

**Strains and maintenance conditions of *Caenorhabditis elegans*.** The nematode culture was synchronized with 2% sodium hypochlorite and 5 M sodium hydroxide. In the sub-chronic toxicity assays, eggs resistant to alkaline lysis were collected and transferred to Petri dishes containing only NGM culture medium and *E. coli* (OP50) until reaching the L4 stage. After reaching the L4 stage of development, these worms were transferred to microplates containing M9 liquid medium and subjected to different concentrations of FPCA in the absence of *E. coli*.

For the assays of reproductive toxicity, locomotor toxicity, stress responses, lifespan, and the expression of superoxide dismutase, glutathione S-transferase, and transcription factor DAF-16, the eggs resistant to alkaline lysis were collected and transferred to Petri dishes containing NGM culture medium, *E. coli* (OP50) and FPCA concentrations (250, 500, or 1000 μg/mL) or water (control) until they reached the L4 stage. When the worms reach the L4 stage of development in the tests of reproductive and locomotor toxicity, thermal stress, and lifespan, they continue to be maintained in plates containing NGM solid medium, *E. coli*, and different concentrations of FPCA or water (control). However, for oxidative stress tests, SOD-3, GST-4, and DAF-16 expression, when they reach the L4 phase of development, the worms are transferred to an M9 liquid medium in the absence of *E. coli*, with different concentrations of FPCA.

**Sub-chronic toxicity.** In this assay, we evaluated the toxic effect of sub-chronic exposure to FPCA on N2 worms. For this, an average of 10 synchronized L4 stage worms were transferred to 96-well microplates containing M9 culture medium (100 μL), in the absence of *E. coli* and FPCA (100 μL) at different concentrations (10–1000 μg/mL). Subsequently, the worms were incubated at 20°C for 24 and 48 hours. As a negative control, the worms were incubated with an M9 culture medium only (200 μL). After the incubation period, worm viability was assessed by touch sensitivity using a platinum wire. Three independent experiments were performed in triplicates.

**Reproductive toxicity.** To assess reproductive toxicity, we analyzed the effects of FPCA on the reproductive capacity of worms. For this, the number of viable progeny was quantified during a five-day reproductive period. In this assay, after synchronization, 5 L4 stage worms pre-treated with water (negative control) or FPCA at concentrations of 250, 500, or 1000 μg/mL were transferred daily to new plates containing NGM/*E. coli* (OP50) medium and water or FPCA at the different experimental concentrations. The number of progeny was evaluated on

each plate after reaching the L3 or L4 larval stage. The results are expressed as the average of three independent experiments.

**Locomotor toxicity.** The effect of FPCA on the locomotor toxicity of N2 nematodes was evaluated in two phases of the nematode life cycle (S1 Fig). The first was the adult phase, corresponding to the period from egg until the second day of L4, and the second was the ageing phase, which went from the L4 stage until the seventh day of life. For this purpose, after synchronization, an average of 10 nematodes in the L4 stage were transferred daily to new Petri dishes containing the treatments with water (negative control) or FPCA (250, 500, or 1000 μg/mL) until they reached the adult and ageing phases. After these periods, the nematodes were transferred to new Petri dishes containing only NGM culture medium, followed by acclimation for 1 min and subsequent evaluation. In the evaluations, the number of sinusoidal bends performed in the 30-s locomotion period was counted. Three independent assays were performed, each in triplicate with 10 nematodes per group.

**Protection against heat stress.** In the heat stress protection assays, an average of 20 L4 stage worms pre-treated for 30 minutes with water (negative control) or FPCA at concentrations of 250, 500, or 1000 μg/mL were transferred to new plates containing NGM/ inactivated by kanamycin *E. coli* (OP50) medium and water or FPCA (250, 500, or 1000 μg/mL), respectively. Heat stress was induced by increasing the culturing temperature from 20˚C to 37˚C, and assessed every hour of exposure during the 6-hour experimental period. The viability of worms exposed to 37˚C at different incubation periods was confirmed after a recovery period of 16 hours at 20˚C, using touch sensitivity with a platinum wire. Three independent experiments were performed in triplicates.

**Protection against oxidative stress.** The assay for protection against oxidative stress was performed by exposing the worms to the oxidizing agent Juglone (5-hydroxy-1,4-naphthoquinone) at a lethal concentration of 250 μM. After synchronization, an average of 10 L4 stage worms pre-treated for 30 minutes with water (control) or experimental concentrations of FPCA (250, 500, or 1000 μg/mL) were transferred to 96-well microplates containing 100 μL of M9 culture medium, 100 μL of FPCA (250, 500, or 1000 μg/mL), and 50 μL of Juglone. As controls, worms pre-incubated with water were exposed to either 250 μL of M9 culture medium (negative control) or 200 μL of M9 medium plus 50 μL of Juglone (positive control). All microplates were incubated at 20˚C, and worm viability was assessed every hour during the 6-hour experimental period. Worm viability was confirmed using touch sensitivity with a platinum wire. Three independent experiments were performed in triplicates.

**Lifespan.** In the lifespan assays, N2 nematodes in the L4 stage were used. On the first day of the L4 stage (day 1), 20 nematodes per group were transferred to new Petri dishes containing NGM + *E. coli* OP50 with water (negative control) or FPCA (250, 500, or 1000 μg/mL). During the first 6 days, corresponding to the reproduction period, the nematodes were transferred daily to new NGM dishes containing the respective treatments. From the seventh day (day 7) on, transfers to new Petri dishes occurred every 2 days. The evaluations consisted of classifying the nematodes as dead or alive until the day the last nematodes died. Nematodes were considered dead when they did not move with or without stimulation by a platinum wire. Nematodes with eggs hatched internally or not visualized in the Petri dishes had their data excluded. Two independent assays were performed in triplicate.

**Expression of SOD-3 and GST-4.** To analyze the expression of the antioxidant enzymes superoxide dismutase (SOD-3) and glutathione-S-transferase (GST-4), CF1553 and CL2166 strains marked with GFP were used. After synchronization, 5 L4 stage worms pre-treated with water (negative control) or concentrations of FPCA (250, 500, or 1000 μg/mL) for 30 minutes were immediately transferred to microscope slides containing 1 mM levamisole as an anesthetic. Subsequently, individual worm images were captured using an epifluorescence

microscope (Nikon Eclipse 50i) connected to a digital camera (Samsung ST64). Images of 5 worms per group were expressed as the average pixel intensity, and the relative fluorescence of the whole body was determined using ImageJ software. Three independent experiments were performed in triplicates.

**DAF-16 translocation.**   To evaluate the translocation of the transcription factor DAF-16, we used the transgenic strain TJ356 with a fusion of the reporter gene daf-16::GFP, which allows visualization of the cellular localization of DAF-16. In this assay, after synchronization, 30 L4 stage worms pre-treated with water (negative control) or concentrations of FPCA (250, 500, or 1000 μg/mL) for 30 minutes were immediately transferred to microscope slides. To monitor the nuclear translocation of DAF-16-GFP, worm images were captured using an epi-fluorescence microscope (Nikon Eclipse 50i) connected to a digital camera (Samsung ST64). The worm images were classified based on the localization of GFP. Thirty animals per group were analyzed, and three independent experiments were performed.

## Statistical analysis

GraphPad Prism 5.1 software (San Diego, CA, USA) was used to perform the statistical analyses. The data are expressed as the mean ± standard error of the mean (SEM). Significant differences between groups were determined using Student's *t*-test for comparison between two groups and analysis of variance (ANOVA) followed by Dunnett's test for comparison between two or more groups. The lifespan assays are represented by the Kaplan-Meier curve, and the *P* values were calculated by the log-rank test. The results were considered significant when *P* <0.05.

## Results

### Identification of the constituents by LC-DAD-MS

The constituents from *C. adamantium* fruits pulp (FPCA) were identified by LC-DAD-MS, using UV, accurate mass, and MS/MS data. The spectral data were compared to data reported in the literature and some compounds were confirmed by injection of authentic standards (Fig 1 and Table 1).

The peaks 1 and 2 revealed the deprotonated ions at *m/z* 165.0415, 179.0574 and 191.0212, which are putatively identified as pentonic acid, hexose, and citric acid. The compounds 3 and 4 showed a band near 280 nm in the UV spectra. Their deprotonated ions (*m/z* 289.0739 and 577.1389) confirmed the molecular formulae $C_{15}H_{14}O_6$ and $C_{30}H_{26}O_{12}$, and these data suggested flavan-3-ol compounds and 4 a dimeric [20]. From *m/z* 577, the product ions *m/z* 407 and 289 confirmed the linkage of two units of procyanidin (B-type). The fragment *m/z* 407 is yielded from retro Diels-Alder fission and subsequently loss of a water molecule, confirming two hydroxyl substituents in the B ring of procyanidin (catechin/epicatechin) [14] and thus it was identified as procyanidin dimer. Besides, compound 3 was identified and confirmed by injection of standard catechin. The compounds 3 and 4 have been described from *C. adamantium* leaves [14].

The compound 5 revealed an intense ion at *m/z* 453.1063 indicating $C_{20}H_{22}O_{12}$. The fragment ions *m/z* 313 are yielded by loss of a hydroxy-methoxy phenyl, while *m/z* 169 is relative to gallic acid from losses of a hydroxy-methoxy phenyl and a hexose. These data are compatible with hydroxy methoxy-phenyl O-hexosyl gallic acid [21].

The compounds 11–14, 18, 20–21, and 24 showed UV spectra similar to ellagic acid ($\lambda_{max}$ ≈260 and 360 nm). The fragment ions at *m/z* 301 are relative to the ellagic acid molecule, which was yielded from losses of 132 and 146 *u* indicating the substituents pentosyl and deoxyhexosyl [14, 22]. Thus, *O*-pentosyl ellagic acid (11 and 13) and *O*-deoxyhexosyl ellagic acid

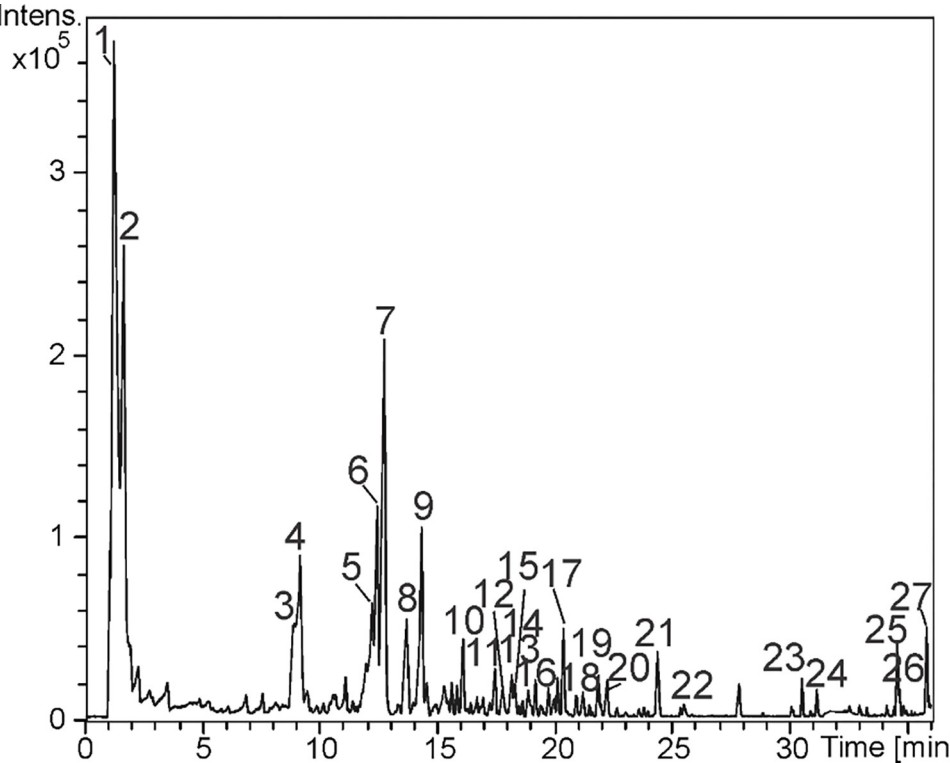

**Fig 1. Base peak chromatogram of *C. adamantium* pulp fruit (FPCA).**

(14) could be identified. These compounds have been identified from *C. adamantium* roots [14]. In addition, the peaks 18, 20, 21, and 24 revealed fragments ions yielded by losses of 15 *u* ($CH_3^\bullet$) from ellagic acid molecule such as the ions *m/z* 300 [*O*-methyl ellagic acid- $CH_3^\bullet$]$^-$ (for 18 and 20), 328 [*O*-trimethyl ellagic- $CH_3^\bullet$]$^-$ (for 21), 313 [*O*-trimethyl ellagic- $2CH_3^\bullet$]$^-$ (for 21 and 24), and 298 [*O*-trimethyl ellagic- $3CH_3^\bullet$]$^-$ (for 21 and 24). Thus, the compounds 18, 20, 21, and 24 were identified as *O*-pentosyl *O-methyl-ellagic* acid, *O*-deoxyhexosyl *O-methyl* ellagic acid, a tri-*O*-methy-ellagic acid derivative and *O*-trimethyl ellagic acid [14, 22]. The compound 12 revealed spectral data compatible with ellagic acid [22], which was also confirmed by the injection of authentic standard.

The compounds 15–17, 19, and 22 showed UV spectra of flavonols ($\lambda_{max} \approx 260$ and 350 nm) [23]. These metabolites showed the same aglycone (m/z 300), which is relative to quercetin and they are yielded by radical losses of hexose, pentose, and deoxyhexose (15–17 and 19). Therefore, 15–17 and 19 were identified as *O*-hexosyl quercetin, *O*-pentosyl quercetin, *O*-pentosyl quercetin and *O*-deoxyhexosyl quercetin, these compounds have been described from *C. adamantium* leaves [14]. In addition, 25 and 26 revealed absorption bands at $\approx 290$ and 335 nm, which indicated flavanones [23]. The deprotonated ions (*m/z* 269.0821 and 269.0827) characterized the molecular formula $C_{16}H_{14}O_4$, and the product ion *m/z* 165, yielded by retro Diels-Alder fission, confirmed the presence of methyl substituent in the A-ring. Thus, the data spectral and the elution profile are compatible with the metabolites 5,7-dihydroxy 6-methylflavanone (25) and 5,7-dihydroxy 8-methylflavanone (26) [14, 24].

**Yield and identification of bioactive compounds.** The yield obtained from the fresh pulp after the lyophilization process was 12.09%. The concentrations of bioactive compounds present in FPCA are shown in Table 2.

**Table 1. Constituents identified in *C. adamantium* pulp fruit (FPCA) by LC-DAD-MS.**

| Peak | RT (min) | Compound | UV (nm) | MF | MS [M-H]⁻ (m/z) | MS/MS (m/z) |
|------|----------|----------|---------|-----|------------------|-------------|
| 1 | 1.2 | Pentonic acid | - | $C_5H_{10}O_6$ | 165.0415 | - |
| | | Hexose | - | $C_6H_{12}O_6$ | 179.0574 | - |
| 2 | 1.6 | Citric acid | - | $C_6H_8O_7$ | 191.0212 | - |
| 3 | 9.0 | Catechin[st] | 282 | $C_{15}H_{14}O_6$ | 289.0739 | 245, 203, 179 |
| 4 | 9.2 | Procyanidin dimer | 281 | $C_{30}H_{26}O_{12}$ | 577.1389 | 407, 289, 245, 203 |
| 5 | 12.2 | Hydroxy methoxy-phenyl O-hexosyl gallic acid | 282 | $C_{20}H_{22}O_{12}$ | 453.1063 | 313, 183, 169 |
| 6 | 12.3 | NI | 280 | $C_{18}H_{26}O_{10}$ | 401.1471 | 245, 221, 203, 191, 177, 164 |
| 7 | 12.7 | NI | 283 | $C_{20}H_{18}O_9$ | 401.0905 | 301, 289, 245 |
| 8 | 13.5 | NI | - | $C_{20}H_{32}O_{10}$ | 431.1948 | 153 |
| 9 | 14.2 | NI | 284, 302[sh] | $C_{25}H_{22}O_{12}$ | 513.1067 | 401, 301, 289, 245, 215 |
| 10 | 15.9 | NI | 282 | $C_{17}H_{30}O_{10}$ | 393.1798 | - |
| 11 | 17.4 | O-pentosyl ellagic acid | 255, 358 | $C_{19}H_{14}O_{12}$ | 433.0438 | 301, 245, 229 |
| 12 | 17.7 | Ellagic acid[st] | 250, 360 | $C_{14}H_6O_8$ | 300.9999 | 283, 245, 229, 201, 173 |
| 13 | 18.1 | O-pentosyl ellagic acid | 252, 360 | $C_{19}H_{14}O_{12}$ | 433.0436 | 301, 229 |
| 14 | 18.3 | O-deoxyhexosyl ellagic acid | 272, 360 | $C_{20}H_{16}O_{12}$ | 447.0594 | 301, 245, 229 |
| 15 | 18.9 | O-hexosyl quercetin | 265, 348 | $C_{21}H_{20}O_{12}$ | 463.0900 | 300, 271, 255, 243 |
| 16 | 19.7 | O-pentosyl quercetin | 265, 355 | $C_{20}H_{18}O_{11}$ | 433.0808 | 300, 271, 255, 243 |
| 17 | 20.3 | O-pentosyl quercetin | 260, 350 | $C_{20}H_{18}O_{11}$ | 433.0798 | 300, 271, 255, 243, 179 |
| 18 | 20.9 | O-pentosyl O-methy-ellagic acid | 251, 352 | $C_{20}H_{16}O_{12}$ | 447.0585 | 315, 300, 271 |
| 19 | 21.1 | O-deoxyhexosyl quercetin | 251, 352 | $C_{21}H_{20}O_{11}$ | 447.0938 | 300, 271, 255, 243, 179 |
| 20 | 22.2 | O-deoxyhexosyl O-methy ellagic acid | 255, 360 | $C_{21}H_{18}O_{12}$ | 461.0745 | 315, 300 |
| 21 | 24.3 | Tri-O-methy-ellagic acid derivative | 270, 360 | $C_{24}H_{24}O_{15}$ | 551.1073 | 343, 328, 313, 298 |
| 22 | 25.5 | Quercetin[st] | 265, 357 | $C_{15}H_{10}O_7$ | 301.0355 | 271, 255, 243, 179, 151 |
| 23 | 30.5 | NI | - | $C_{18}H_{32}O_5$ | 327.2191 | 221, 211, 183, 171 |
| 24 | 31.1 | O-trimethyl ellagic acid | 285, 357 | $C_{17}H_{12}O_8$ | 343.0466 | 313, 298, 270 |
| 25 | 34.5 | 5,7-dihydroxy 6-methylflavanone | 290, 333[sh] | $C_{16}H_{14}O_4$ | 269.0821 | 227, 199, 183, 171, 165 |
| 26 | 34.6 | 5,7-dihydroxy 8-methylflavanone | -294, 336[sh] | $C_{16}H_{14}O_4$ | 269.0827 | 227, 199, 165 |
| 27 | 35.8 | NI | - | $C_{14}H_{20}O_4$ | 251.1288 | 233, 218, 207, 193, 167 |

NI: non identified; RT: retention time; MF: molecular formula; [sh]: shoulder; [st]: confirmed by injection of authentic standard. All the molecular formulae were determined by accurate mass considering error and mSigma up to 10 and 30, respectively.

## *In vitro* antioxidant activity

The evaluation of the *in vitro* antioxidant activity of FPCA, represented by the concentration capable of inhibiting 50% ($IC_{50}$) of DPPH• and ABTS•+ radicals, are shown in Table 3. The

**Table 2. Bioactive compounds quantified in FPCA.**

| Compounds | Results |
|-----------|---------|
| Phenolic compounds | 3972.42 ± 0.93 mg EAG/100 g |
| Flavonoids | 85.13 ± 0.37 mg QE/100 g |
| β-Carotene | 0.062 ± 0.014 mg/g |
| Lycopene | 0.029 ± 0.010 mg/g |
| Chlorophyll *a* | 0.113 ± 0.02 µg/g |
| Chlorophyll *b* | 0.077 ± 0.031 µg/g |
| Ascorbic acid | 1454.46 ± 27.17 mg/100 g |

Values are expressed as the mean ± SEM.

**Table 3. Antioxidant activity of *C. adamantium* fruit pulp (FPCA).**

| Samples | DPPH$^{\bullet}$ | ABTS$^{\bullet+}$ |
|---|---|---|
| | IC$_{50}$ (µg/mL) | IC$_{50}$ (µg/mL) |
| Ascorbic acid | 2.65 ± 0.20 | 1.43 ± 0.09 |
| BHT | 14.58 ± 2.15 | 10.15 ± 0.94 |
| FPCA | 210.5 ± 28.0 | 89.12 ± 0.03 |

Values are expressed as the mean ± SEM.

FPCA was more efficient in scavenging the ABTS$^{\bullet+}$ radical than the DPPH$^{\bullet}$ radical, with an IC$_{50}$ approximately 2.36 times lower.

## *In vivo* assays

**Sub-chronic toxicity.** Initially, the sub-chronic toxicity of different FPCA concentrations (0.01–1 mg/mL) was evaluated *in vivo*. Fig 2A and 2B, respectively, show that at none of the concentrations evaluated did FPCA promote toxicological changes, represented by the viability of the nematodes after 24 and 48 h. From these results, we could safely define the concentrations for the next assays.

**Reproductive toxicity.** The effect of FPCA on the number of viable progenies of N2 nematodes is an indicator of reproductive toxicity. Fig 3A shows that none of the FPCA concentrations evaluated promoted changes in the daily or total number of viable progenies (Fig 3B). These results indicate that different concentrations of FPCA do not promote toxic effects that impair the physiological patterns of nematode reproductive capacity.

**Locomotor toxicity.** The effect of FPCA on the locomotor capacity of nematodes up to the young adult and ageing phases is an important toxicity parameter at different stages of the life cycle. The results show that FPCA did not promote a decline or improvement in nematode motility in the young adult phase (Fig 4A). On the other hand, a significant improvement in the motility of the nematodes treated with the different concentrations of FPCA was observed during the ageing phase (Fig 4B). At this stage of the life cycle, the body bending frequency of the nematodes in the control group was 9.95 ± 0.45, whereas in the nematodes treated with FPCA it was 11.30 ± 0.27 (250 µg/mL), 12.05 ± 0.35 (500 µg/mL), and 13.50 ± 0.40 (1000 µg/mL). These are improvements of 13.56%, 21.10%, and 35.67% in the number of body bends compared to the control group value.

**Protection against heat stress.** The thermal stress protection assay showed the protective effect of FPCA on nematode viability during a 6-hour period (Fig 5). In the first hour of exposure to heat stress, the control group had 76.32 ± 3.89% viable nematodes, while the nematodes treated with FPCA at 250, 500, and 1000 µg/mL had viabilities of 93.75 ± 1.83%, 89.46 ± 3.57% and 90.00 ± 5.00%, respectively. At the end of the experimental period (6 h), the control group showed only 2.50 ± 1.33% viable nematodes, while the nematodes treated with FPCA 250, 500, and 1000 µg/mL had viabilities of 13.89 ± 1.62%, 13.42 ± 4.40%, and 23.33 ± 1.66%, respectively.

**Protection against oxidative stress.** The increase in resistance to oxidative stress demonstrates a beneficial protective effect against the stressor Juglone, a powerful generator of reactive oxygen species. In the oxidative stress protection assay, the nematodes treated with FPCA resisted the action of the chemical oxidizing agent throughout the evaluation period (Fig 6).

**Lifespan.** To demonstrate the ability of FPCA to prolong life, we evaluated its effects on the average and maximum lifespan of wild-type N2 nematodes. The results show that FPCA increased the average and maximum lifespan of the nematodes in a dose-dependent manner

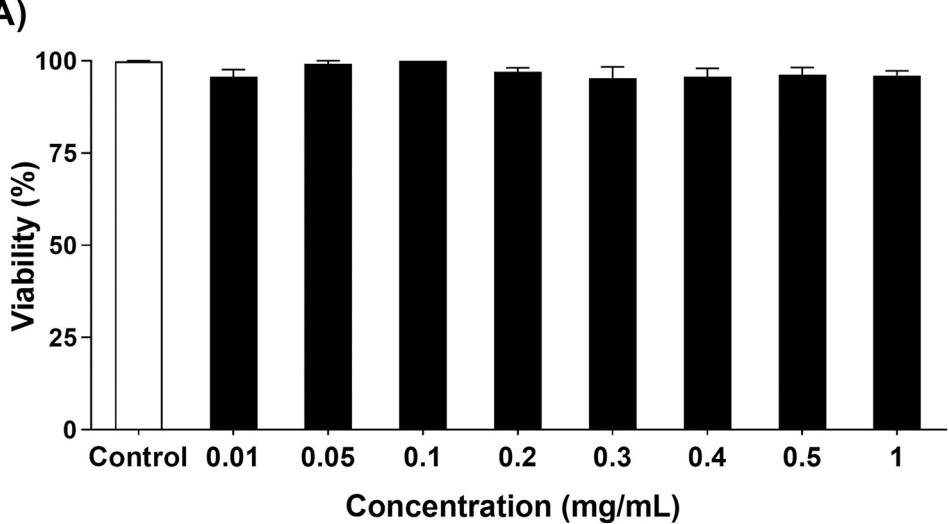

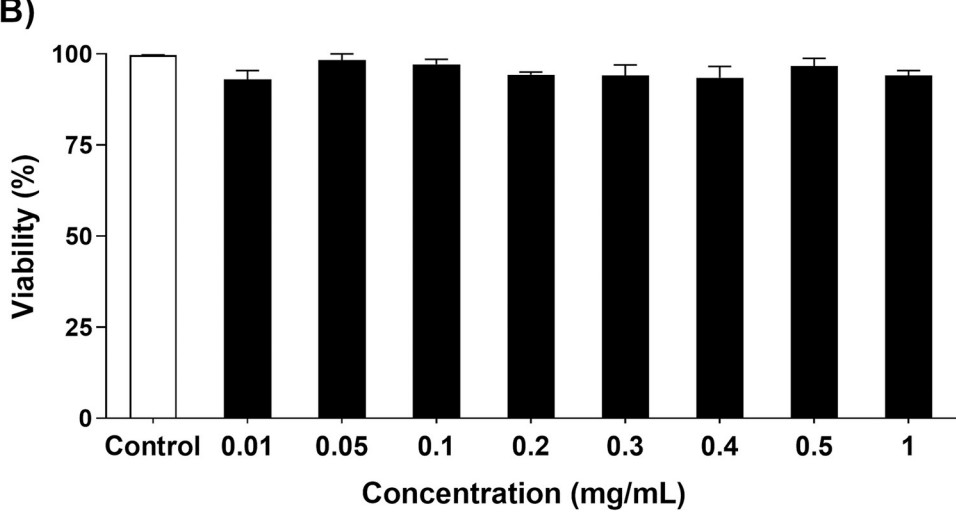

**Fig 2.** Sub-chronic toxicity of the *C. adamantium* fruit pulp (FPCA) in *C. elegans* N2 after: (A) 24 h and (B) 48 h. Values are expressed as the mean ± SEM ($n = 3$). * $P < 0.05$, treated group vs. control group (M9).

(Fig 7 and Table 4). The average lifespan of the nematodes treated with FPCA was extended by 3.5 days (250 μg/mL), 4.5 days (500 μg/mL), and 4.5 days (1000 μg/mL). The effects of FPCA on maximum lifespan were even greater, prolonging the life of the nematodes by 5.5 days (250 μg/mL), 7.5 days (500 μg/mL), and 8.5 days (1000 μg/mL).

**Expression of SOD-3 and GST-4.** The ability of FPCA to modulate target genes related to the endogenous antioxidant defence system was observed in the transgenic strains CF1553

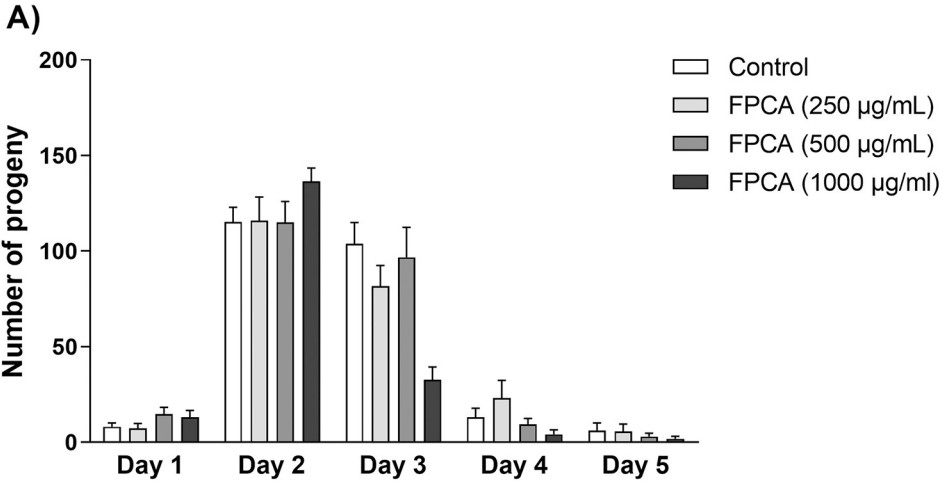

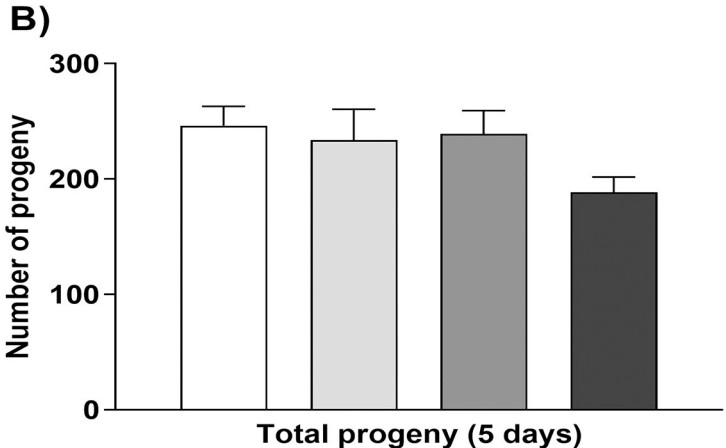

**Fig 3. Effect of the *C. adamantium* fruit pulp (FPCA) on reproductive capacity in *C. elegans* N2.** (A) Daily number of progeny and (B) total number of progeny in 5 days. Values are expressed as the mean ± SEM. * $P < 0.05$, treated group vs. control group.

(SOD-3:: GFP) and CL2166 (GST- 4:: GFP). The results showed significant increases in SOD-3 fluorescence of 6.33, 49.33, and 54.67% in nematodes treated with FPCA at 250, 500, and 1000 μg/mL, respectively (Fig 8). In addition, FPCA (1000 μg/ml) increased GST-4 expression by 48.66% (Fig 9).

**Subcellular localization of the DAF-16 transcription factor.** DAF-16/FOXO is one of the main transcription factors involved in the regulation of genes related to the antioxidant defence system and longevity. In cells under basal stress, DAF-16/FOXO remains inactive in the cytoplasmic region. To demonstrate the involvement of FPCA in the activation of this pathway, we evaluated the subcellular localization of the DAF-16 transcription factor. The results show that all FPCA concentrations induced DAF-16 translocation to the intermediary and nuclear regions of the cells (Fig 10A and 10B). FPCA induced greater translocations to the intermediary region of the cells; thus, the nematodes treated with FPCA (250, 500, and 1000 μg/mL) showed intermediary translocations of 91.67 ± 4.91, 84.50 ± 0.87, and

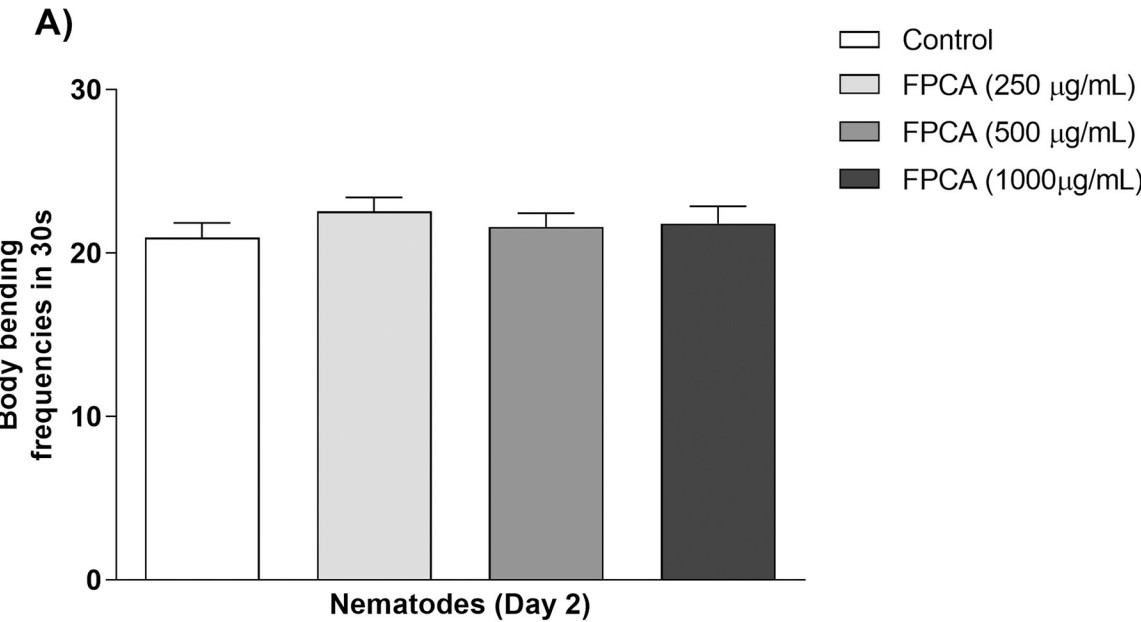

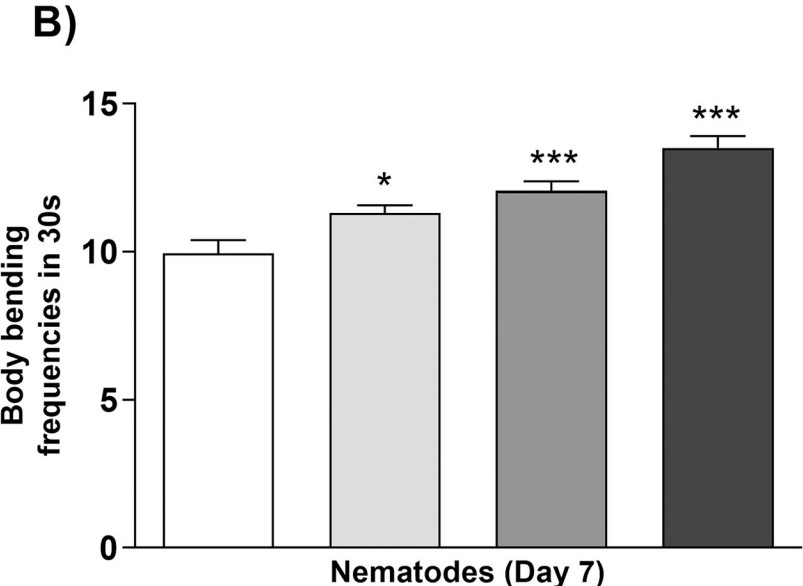

**Fig 4.** Effects of *C. adamantium* fruit pulp (FPCA) on the locomotion of *C. elegans* N2 in the: (A) young adult phase and (B) ageing phase. Values are expressed as the mean ± SEM. * $P < 0.05$ and *** $P < 0.001$, treated group vs. control group.

79.50 ± 2.02%, respectively, while the control group exhibited 55.00 ± 8.66%. However, translocation to the nuclear region occurred only in the nematodes treated with FPCA, corresponding to an increase of 4.67 ± 2.40% (250 µg/mL), 14.17 ± 0.93% (500 µg/mL), and 16.50 ± 3.62% (1000 µg/mL).

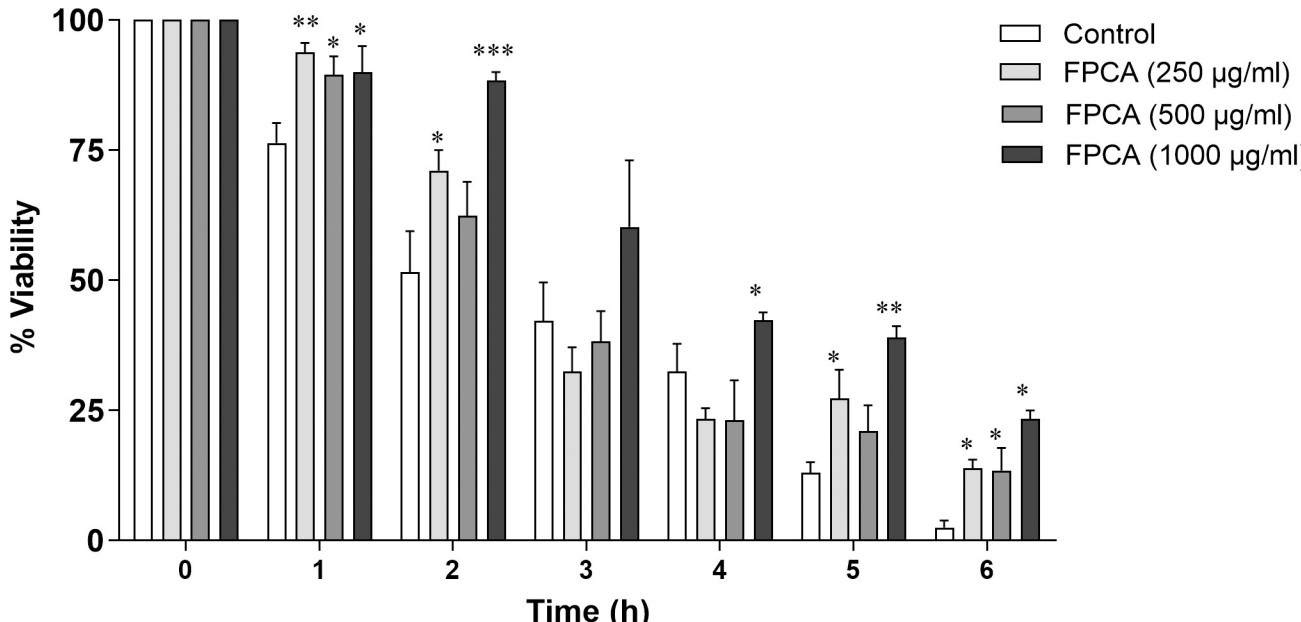

**Fig 5. Protective effect of *C. adamantium* fruit pulp (FPCA) on *C. elegans* N2 exposed to heat stress.** Values are expressed as the mean ± SEM. * $P < 0.05$; ** $P < 0.01$; *** $P < 0.001$, treated group vs. control group (Juglone).

## Discussion

The Brazilian Cerrado biome is home to different fruit species with unique organoleptic characteristics, reflecting the diversity of bioactive compounds and their potential for the development of nutraceutical foods. In this context, native fruits stand out because they are considered natural sources of bioactive substances derived from secondary metabolites, such as alkaloids,

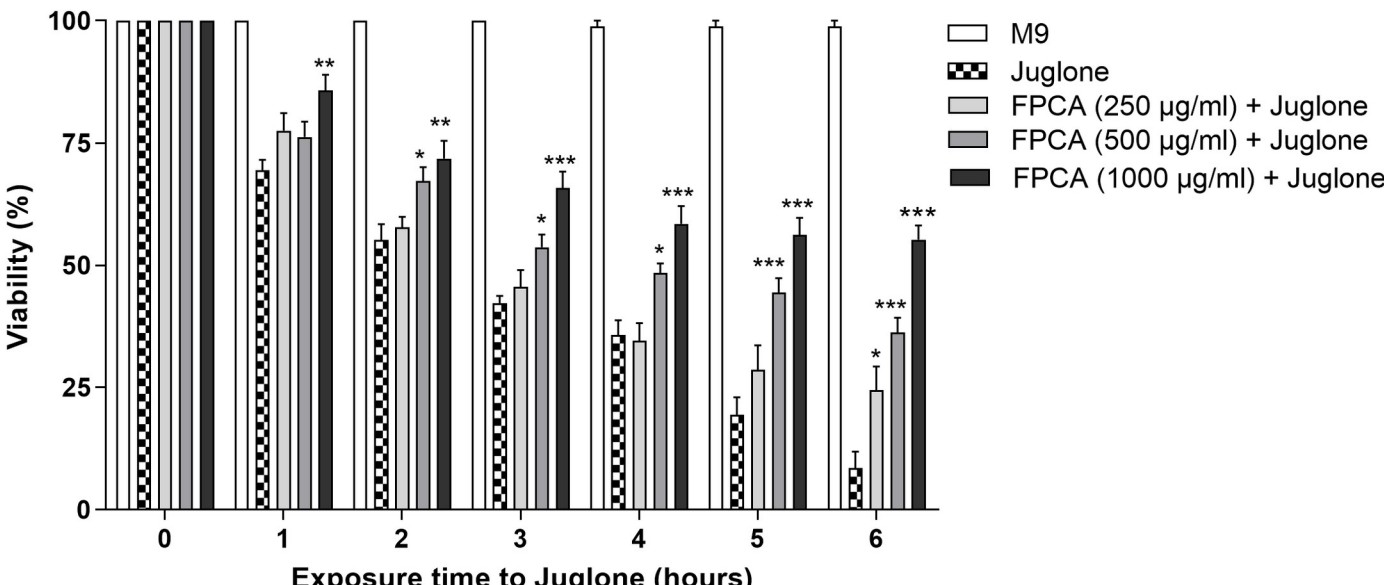

**Fig 6. Protective effect of the *C. adamantium* fruit pulp (FPCA) in *C. elegans* N2 exposed to oxidative stress induced by Juglone.** Values are expressed as the mean ± SEM. * $P < 0.05$; ** $P < 0.01$; *** $P < 0.001$, treated group vs. control group.

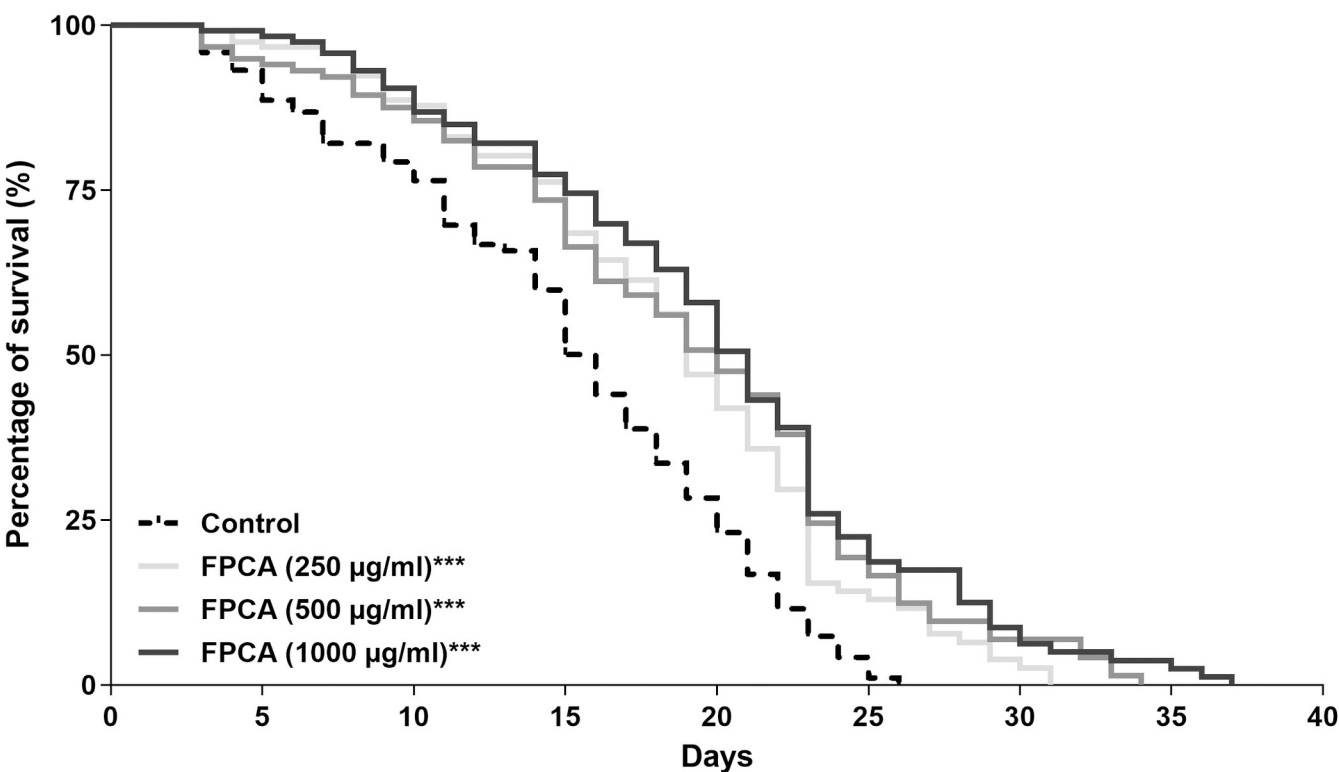

**Fig 7. Lifespan of *C. elegans* N2 treated with *C. adamantium* fruit pulp (FPCA).** ***Statistically significant results ($P < 0.0001$), treated group vs. control group.

glycosides, fatty acids, terpenoids, and polyphenols [25] The beneficial properties of different native fruits are associated with their chemical constituents that have relevant biological activities, such as antimicrobial [3, 24, 26], anti-proliferative [27, 28], anti-inflammatory [29, 30], and antioxidant activities [31]. Among the native fruit species, we investigated the chemical constituents and biological properties of the *Campomanesia adamantium* fruit pulp (FPCA). The presence of phenolic compounds and ascorbic acid has also been verified in another species of the genus *Camponamesia*, *C. rufa*, and linked to the antioxidant activity observed by Abreu et al. [32].

In this study, we identified in the FPCA chemical constituents belonging to the class of phenolic compounds, including phenolic acids (gallic acid and ellagic acid) and flavonoids (catechin, epicatechin, quercetin, and methylflavan). Organic carboxylic acids (pentanoic acid and citric acid) and monosaccharide hexose were also identified, along with ascorbic acid and lipophilic pigments, such as β-carotene, lycopene, and chlorophylls *a* and *b*. Phenolic compounds are described as the main antioxidant bioactive compounds present in plants, and they can

**Table 4. Effects of treatments with FPCA on the lifespan of N2 nematodes.**

| Treatment (µg/ mL) | Mean lifespan (Days) | Mean extension (%) | Maximum lifespan (Days) | Maximum extension (%) | Log–rank Test vs. Control | Total number of nematodes |
|---|---|---|---|---|---|---|
| Control | 16.00 ± 1.00 | - | 25.50 ± 0.05 | - | - | 120 |
| FPCA (250) | 19.50 ± 1.50 | 21.87 | 31.00 ± 0.00 | 21.56 | <0.0001*** | 120 |
| FPCA (500) | 20.50 ± 1.50 | 28.12 | 33.00 ± 1.00 | 29.41 | <0.0001*** | 120 |
| FPCA (1000) | 20.50 ± 0.50 | 28.12 | 34.00 ± 3.00 | 33.33 | <0.0001*** | 120 |

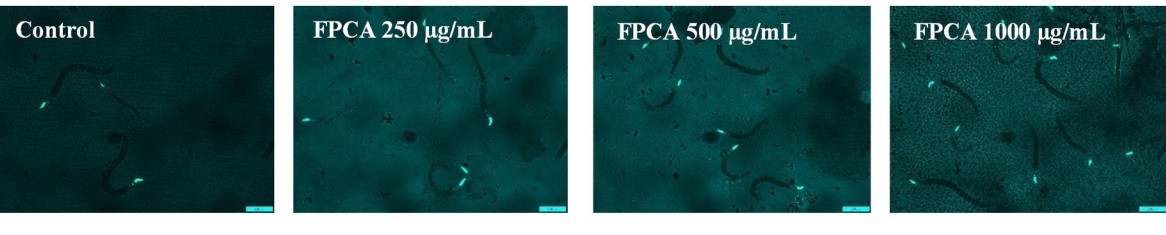

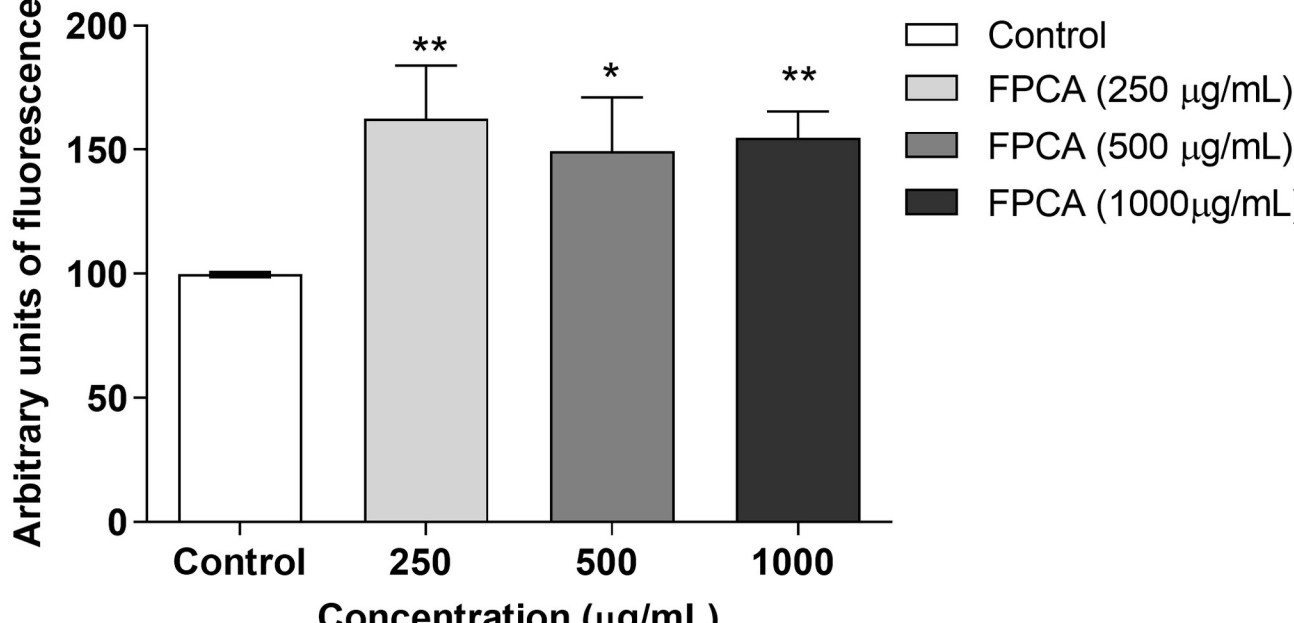

**Fig 8. Expression of SOD-3::GFP in nematodes (CF1553 [*sod-3p*:GFP]) treated with *C. adamantium* fruit pulp (FPCA).** Values are expressed as the mean ± SEM. * $P < 0.05$ and ** $P < 0.01$, treated group vs. control group.

eliminate free radicals and protect cellular constituents against oxidative damage [33]. Among these, flavonoids act through different mechanisms, such as via direct elimination of reactive oxygen species, chelation of metals, and activation of antioxidant enzymes [34]. Intermediate compounds of pentanoic acid are involved in cellular defence mechanisms, inactivating the enzyme neuronal nitric oxide synthase via oxidative demethylation, preventing nitric oxide from reacting with the superoxide anion radical and forming peroxynitrite, which at high levels is associated with the pathogenesis of neurodegenerative diseases [35, 36]. Citric acid, in addition to being an intermediate agent of the tricarboxylic acid cycle in the metabolism of aerobic organisms, is widely used in the food and pharmaceutical industry due to its buffering, anticoagulant, anti-inflammatory, and antioxidant properties [37].

Ascorbic acid, known as vitamin C, is considered an essential micronutrient and is present in vegetables and fruits [38]. This compound performs important functions in numerous physiological processes, acting as a reducing agent in most reactions involving reactive oxygen and nitrogen species and acting as an enzymatic cofactor of the main antioxidant enzymes superoxide dismutase, catalase, and glutathione [39, 40]. The ingestion of ascorbic acid at physiological concentrations is associated with the prevention of heart disease, anti-inflammatory activity, collagen biosynthesis, antioxidant protection against UV rays [39], and increased

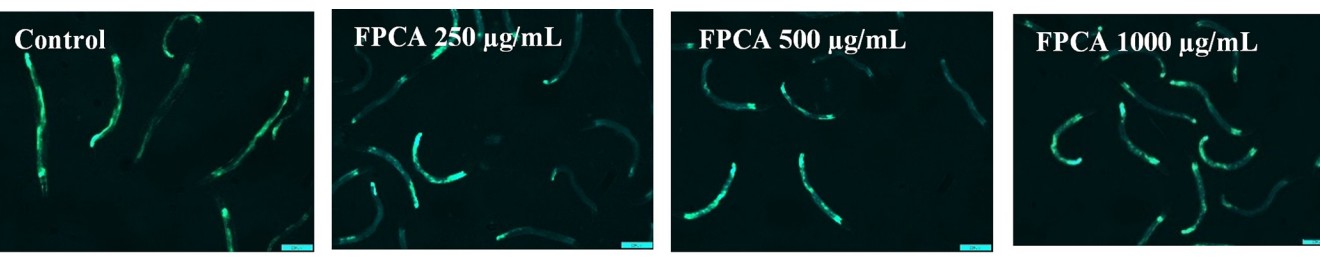

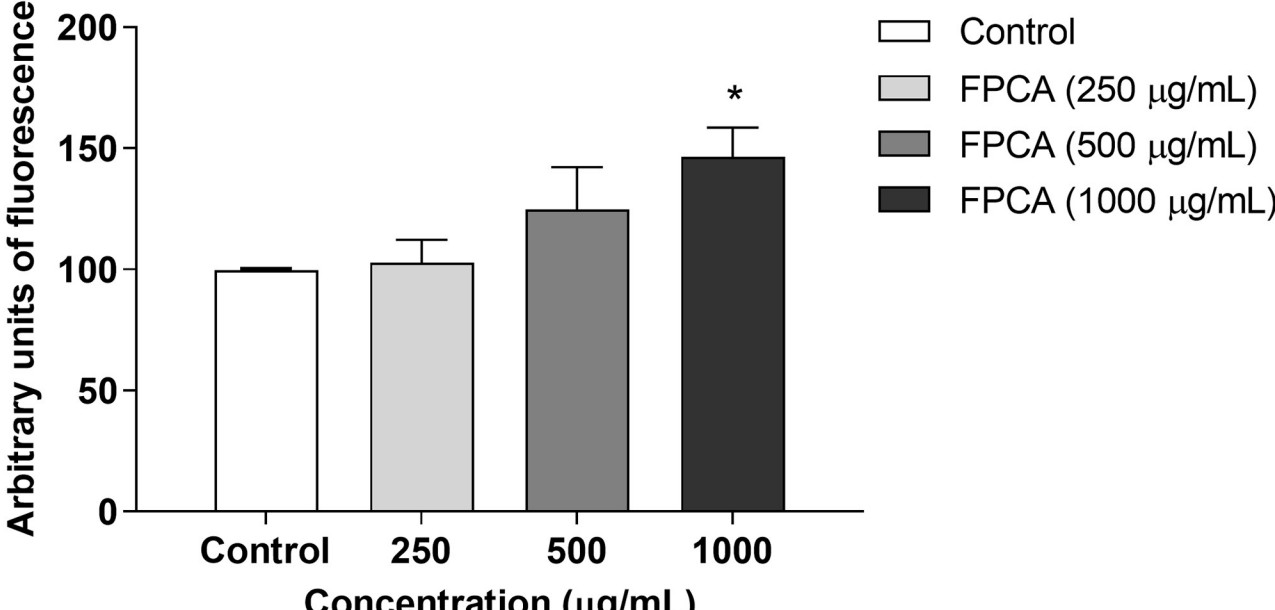

**Fig 9. Expression of GST-4::GFP in nematodes (CL 2166 [*gst-4p*: GFP]) treated with *C. adamantium* fruit pulp (FPCA).** Values are expressed as the mean ± SEM. * $P < 0.05$, treated group vs. control group.

lifespan in mice [41] and in *C. elegans* [42]. Other studies have also identified phenolic compounds, flavonoids, and ascorbic acid in the *C. adamantium* fruit extract and related them to its antimicrobial and antioxidant properties [24, 43].

Carotenoids and chlorophyll pigments are described for their antioxidant properties and are associated with the prevention of chronic diseases [44, 45]. Although humans and other animals cannot synthesize carotenoids, these compounds have important biological activities in reproduction, embryonic development, immune modulation, and ocular tissue maintenance [46]. Chlorophyll, the main pigment of plants, has lipophilic characteristics and antimutagenic and antioxidant properties [47, 48]. Thus, the antioxidant activity of FPCA demonstrated by the direct scavenging of radicals can be attributed to the isolated and/or combined effect of its chemical compounds, since they can act by different antioxidant mechanisms, including promoting the neutralization of free radicals through donation of hydrogen atoms and/or sequestering electrons from unstable molecules.

In recent decades, there has been a growing interest in natural antioxidant compounds that have beneficial effects capable of promoting a better quality of life and healthy ageing [49–51]. For this purpose, fruits stand out because they are already part of the human diet. However, to

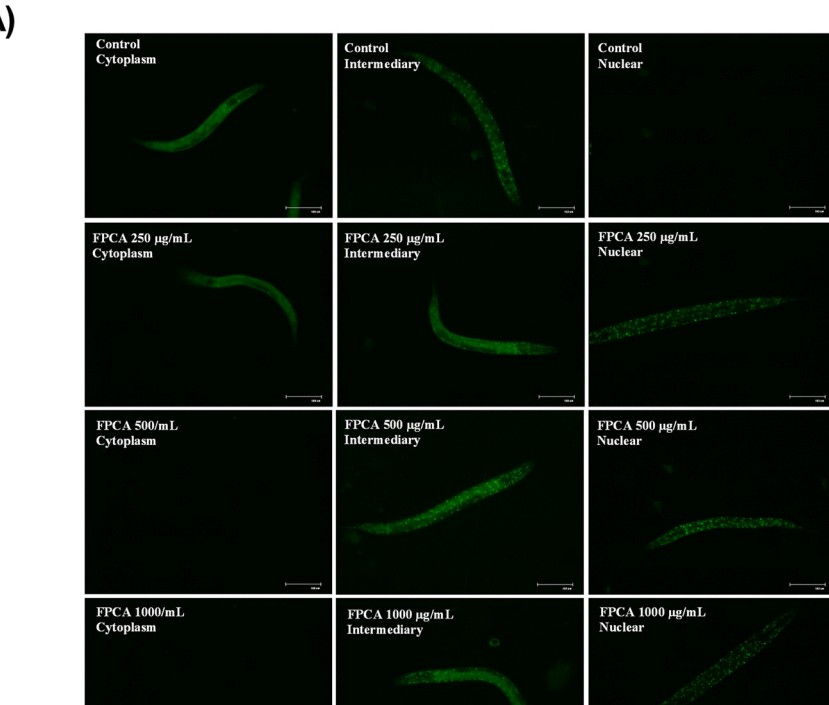

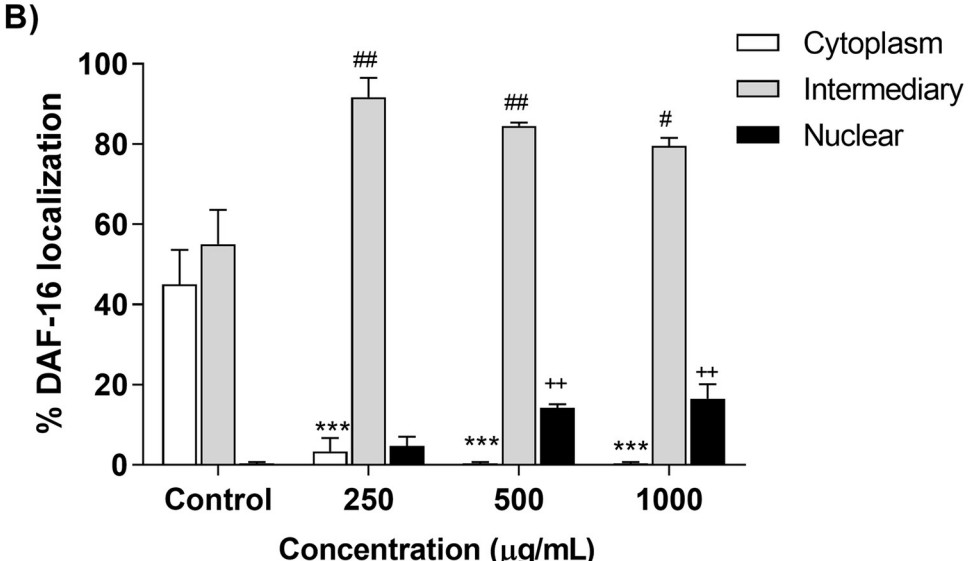

**Fig 10.** (A) Expression and (B) subcellular localization of DAF-16 in nematodes (TJ356 [daf-16p: daf-16a/b: GFP + role-6 (su1006)]) treated with *C. adamantium* fruit pulp (FPCA). Values are expressed as the mean ± SEM. *** $P < 0.001$, cytoplasmic localization when compared to the control group. # $P < 0.05$ and ## $P < 0.01$, intermediary localization when compared to the control group. ++ $P < 0.01$, nuclear localization when compared to the control group.

ensure their efficacy and safe consumption, toxicological evaluations and confirmation of their biological properties are necessary. From this perspective, the *in vivo* experimental model *C. elegans* is an important tool to investigate the biological properties, toxicological effects, and molecular mechanisms of isolated compounds and/or natural products [52].

The toxicological parameters evaluated show that the nematodes exposed to FPCA did not present any impairment in their physiological or viability parameters. In contrast, a protective effect of FPCA was demonstrated in the parameters of locomotor toxicity in middle-aged adult nematodes. In *C. elegans*, muscle cells gradually lose vitality, causing a decline in mobility and physiological changes that are closely related to the effects of ageing [53]. Body movements become sporadic from the sixth to the tenth day of life, but adult nematodes with faster locomotor decline are more likely to have a shorter lifespan [54]. In addition, the absence of changes observed in this study corroborates the study by Viscardi et al. [29], which demonstrated that the peels and seeds of the *C. adamantium* fruit do not have toxic effects in mice.

The beneficial effects of FPCA were demonstrated *in vivo* in antioxidant assays under heat and oxidative stress. When living organisms are exposed to stressors, such as high temperature, the protein denaturation process begins, which affects numerous biomolecules and consequently their structural and metabolic functions [18]. In this study, FPCA's protective activity demonstrated against heat stress may have been related to the presence of chemical constituents identified in FPCA, including the flavonoids epicatechin and catechin and their oligomers, procyanidins, which due to their antioxidant and free radical scavenging properties have shown protective effects against heat stress in *C. elegans* [55].

Oxidative stress is among the main factors that accelerate the ageing process and limit lifespan in both humans and other animals [56]. This study shows that nematodes, when treated with FPCA and exposed to the pro-oxidant agent Juglone, a chemical agent that induces reactive species production [57], were more resistant to oxidative stress, as demonstrated by their greater viability. These data demonstrate the protective effect of FPCA against oxidative stress, which may be related to its antioxidant capacity, involving the activation of direct mechanisms, such as the removal of free radicals, and indirect mechanisms, such as modulation of the endogenous antioxidant system through the expression of antioxidant enzymes that control the levels of reactive oxygen species and reactive nitrogen species [58]. Other signalling pathways may also be involved in this process, given the wide variety of chemical constituents identified in FPCA that can act both in isolation and synergistically.

In *C. elegans*, resistance to different stresses is related to an increase in the lifespan [59]. This relationship was observed in the present study because, in addition to promoting protective effects against stressors, FPCA increased the average lifespan and prolonged the useful life of *C. elegans*. Blueberry, another fruit rich in bioactive phytochemicals such as proanthocyanidins, also promotes beneficial effects against oxidative stress, improves locomotion, and increases lifespan in nematodes by modulating DAF-16 and upregulating antioxidant gene expression [60]. Cranberry, a fruit rich in phenolic compounds, increases lifespan and promotes resistance to heat stress by modulating the antioxidant pathways DAF-16/FOXO and SKN-1/Nrf-2 [61].

Among the evaluated mechanisms that may help in understanding the beneficial responses promoted by FPCA is the modulation of the expression of target antioxidant genes, such as superoxide dismutase (SOD-3) and glutathione S-transferase (GST-4), and the activation of the transcription factor DAF-16 [62]. According to our results, FPCA activated and induced the translocation of DAF-16 to the nucleus and modulated the expression of SOD-3 and GST-4. The activation of endogenous antioxidant pathways is important for detoxification in *C. elegans* because these induce mechanisms of protection against damage caused by stress and promote longevity [63]. In humans, the ageing process is associated with functional and

morphological changes that lead to the progressive decline of biological functions. Interventions, especially dietary interventions, that promote beneficial effects and positive impacts during the ageing phase can prolong lifespan and promote health.

## Conclusion

In conclusion, our results show that FPCA has a wide variety of chemical compounds and no toxicity. It has a protective effect against heat and oxidative stress and increases the lifespan of *C. elegans* by directly scavenging free radicals, increasing the expression of the antioxidant enzymes superoxide dismutase and glutathione S-transferase, and activating the transcription factor DAF-16. These findings demonstrate the functional potential of the *C. adamantium* fruit species native to the Brazilian Cerrado biome for the control of oxidative stress, with a perspective for the development of new products to promote a healthy lifespan and prevent related diseases.

## Supporting information

**S1 Fig. Diagram of the treatments with the *C. adamantium* fruit pulp (FPCA) during the different phases of the life cycle of *C. elegans*.**
(DOC)

**S2 Fig.**
(TIF)

## Acknowledgments

The authors would like to thank the government agencies that support science in Brazil: Fundação de Apoio ao Desenvolvimento da Educação, Ciência e Tecnologia do Estado de Mato Grosso do Sul (FUNDECT); Coordenação de Aperfeiçoamento de Pessoal de Nível Superior (CAPES); Conselho Nacional de Desenvolvimento Científico e Tecnológico (CNPq), Fundação de Apoio à Pesquisa no estado do Rio Grande do Sul (FAPERGS) e Financiador de Estudos e Projetos (FINEP).

## Author Contributions

**Conceptualization:** Laura Costa Alves de Araújo, Denise Brentan da Silva, Kely de Picoli Souza, Edson Lucas dos Santos.

**Data curation:** Daiana Silva Ávila, Kely de Picoli Souza, Edson Lucas dos Santos.

**Formal analysis:** Laura Costa Alves de Araújo, Danielle Araujo Agarrayua, Daiana Silva Ávila, Denise Brentan da Silva, Carlos Alexandre Carollo, Jaqueline Ferreira Campos, Kely de Picoli Souza, Edson Lucas dos Santos.

**Funding acquisition:** Kely de Picoli Souza, Edson Lucas dos Santos.

**Investigation:** Laura Costa Alves de Araújo, Natasha Rios Leite, Paola dos Santos da Rocha, Debora da Silva Baldivia, Danielle Araujo Agarrayua, Daiana Silva Ávila, Denise Brentan da Silva, Carlos Alexandre Carollo, Jaqueline Ferreira Campos.

**Methodology:** Laura Costa Alves de Araújo, Natasha Rios Leite, Paola dos Santos da Rocha, Debora da Silva Baldivia, Danielle Araujo Agarrayua, Daiana Silva Ávila, Denise Brentan da Silva, Carlos Alexandre Carollo, Jaqueline Ferreira Campos.

**Project administration:** Kely de Picoli Souza, Edson Lucas dos Santos.

**Resources:** Laura Costa Alves de Araújo, Paola dos Santos da Rocha, Kely de Picoli Souza, Edson Lucas dos Santos.

**Software:** Laura Costa Alves de Araújo, Natasha Rios Leite, Paola dos Santos da Rocha, Danielle Araujo Agarrayua, Daiana Silva Ávila, Denise Brentan da Silva, Carlos Alexandre Carollo, Jaqueline Ferreira Campos, Kely de Picoli Souza, Edson Lucas dos Santos.

**Supervision:** Kely de Picoli Souza, Edson Lucas dos Santos.

**Validation:** Paola dos Santos da Rocha, Debora da Silva Baldivia, Denise Brentan da Silva, Carlos Alexandre Carollo, Jaqueline Ferreira Campos, Kely de Picoli Souza, Edson Lucas dos Santos.

**Visualization:** Laura Costa Alves de Araújo, Paola dos Santos da Rocha, Kely de Picoli Souza, Edson Lucas dos Santos.

**Writing – original draft:** Laura Costa Alves de Araújo, Denise Brentan da Silva, Carlos Alexandre Carollo, Kely de Picoli Souza, Edson Lucas dos Santos.

**Writing – review & editing:** Paola dos Santos da Rocha, Debora da Silva Baldivia, Denise Brentan da Silva, Jaqueline Ferreira Campos, Kely de Picoli Souza, Edson Lucas dos Santos.

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
