## [Editor Report · Decision Letter 0]

20 Jun 2023

PONE-D-23-17391*Campomanesia adamantium* O Berg. fruit, native to Brazil, can protect against oxidative stress and promote longevityPLOS ONE

Dear Dr. dos Santos,

Thank you for submitting your manuscript to PLOS ONE. After careful consideration, we feel that it has merit but does not fully meet PLOS ONE’s publication criteria as it currently stands. Therefore, we invite you to submit a revised version of the manuscript that addresses the points raised during the review process.

 Please address the issue and submit revised manuscript. 

We look forward to receiving your revised manuscript.

Kind regards,

Manoj Kumar

Academic Editor

PLOS ONE

Journal Requirements:

    "This work was supported by grants from Fundação de Apoio ao Desenvolvimento da Educação, Ciência e Tecnologia do Estado de Mato Grosso do Sul (FUNDECT); Coordenação de Aperfeiçoamento de Pessoal de Nível Superior (CAPES); Conselho Nacional de Desenvolvimento Científico e Tecnológico (CNPq), Fundação de Apoio à Pesquisa no estado do Rio Grande do Sul (FAPERGS) e Financiador de Estudos e Projetos (FINEP)."

Additional Editor Comments:

Dear Author,

Before proceeding further kindly address following issues:

Material and methods:

Material, chemical and other components used are not mentioned in the material methodology. I must be mentioned separately in a section 2.1. Materials (make should be mentioned).

Authors must add a flow diagram showing methodology followed in the experimentation and also show the analysis performed at each stage. This flow diagram will definitely improve the readability of the manuscript.

I am not able to find "conclusion" in manuscript. Please address the issues.

Kind regards

Manoj Kumar

---

## [Author Response · Author response to Decision Letter 0]

10 Jul 2023

Response to reviewers 

We are very grateful for the comments raised during the review process. All suggestions have been analyzed and the manuscript has been modified. We appreciate the suggestions in order to improve the final quality of the manuscript. Below we present our statements, and we hope they are sufficient for the requirements of PLOS ONE.

Journal Requirements:

Answer: The manuscript has been revised in accordance with the observations to meet the stylistic requirements as publication criteria of PLOS ONE.

Answer: The in vivo studies described in this manuscript were conducted using the Caenorhabditis elegans experimental model. Ethical authorization or approval from ethics commissions or committees is not required for experimentation with this nematode.

Answer: We have reviewed the provided link and identified an error in completing the form. However, all the research data referenced in the manuscript are readily available.

 "This work was supported by grants from Fundação de Apoio ao Desenvolvimento da Educação, Ciência e Tecnologia do Estado de Mato Grosso do Sul (FUNDECT); Coordenação de Aperfeiçoamento de Pessoal de Nível Superior (CAPES); Conselho Nacional de Desenvolvimento Científico e Tecnológico (CNPq), Fundação de Apoio à Pesquisa no estado do Rio Grande do Sul (FAPERGS) e Financiador de Estudos e Projetos (FINEP)."

Answer: The Acknowledgments section was changed in line 617-622 of the revised manuscript, as bellow. 

“The authors would like to thank the government agencies that support science in Brazil: Fundação de Apoio ao Desenvolvimento da Educação, Ciência e Tecnologia do Estado de Mato Grosso do Sul (FUNDECT); Coordenação de Aperfeiçoamento de Pessoal de Nível Superior (CAPES); Conselho Nacional de Desenvolvimento Científico e Tecnológico (CNPq), Fundação de Apoio à Pesquisa no estado do Rio Grande do Sul (FAPERGS) e Financiador de Estudos e Projetos (FINEP).”

Answer: An error occurred while filling out the submission form, and no information is available to be provided to any repository. However, all the research data pertaining to the study can be found in the manuscript.

Answer: 

The Supplementary Material caption was inserted on lines 852-854 of the revised manuscript, as below. 

“Supplementary material

S1 Fig. Diagram of the treatments with the C. adamantium fruit pulp (FPCA) during the different phases of the life cycle of C. elegans.”

Additional Editor Comments:

Before proceeding further kindly address following issues:

Material and methods:

Material, chemical and other components used are not mentioned in the material methodology. I must be mentioned separately in a section 2.1. Materials (make should be mentioned).

Answer: Materials, chemical components, and other components used in the manuscript's methodology were mentioned separately in the Materials section, lines 80-86 of the revised manuscript, as follows. 

“The chemicals were purchased from Sigma-Aldrich: formic acid, 2,2-diphenyl-1-picrylhydrazyl, 2,2'-azinobis-(3-ethylbenzothiazoline-6-sulfonic acid, Juglone (5-hydroxy-1,4-naphthoquinone) 2,6-dichlorophenolindophenol-sodium (DCIP), potassium persulfate, butylated hydroxytoluene (BHT), quercetin, oxalic acid and sodium hypochlorite; Dinâmica: methanol, acetone, hexane, Folin-Ciocalteu, sodium carbonate, aluminum chloride hexahydrate, ascorbic acid and sodium hydroxide; Diversey: Sumaveg®.”

Authors must add a flow diagram showing methodology followed in the experimentation and also show the analysis performed at each stage. This flow diagram will definitely improve the readability of the manuscript.

Answer: The flow diagram was attached in the option “others” in the window "Attach Files".

I am not able to find "conclusion" in manuscript. Please address the issues.

Answer: The conclusion is highlighted on lines 605-611 of the revised manuscript.

---

## [Decision Letter · Decision Letter 1]

22 Aug 2023

PONE-D-23-17391R1*Campomanesia adamantium* O Berg. fruit, native to Brazil, can protect against oxidative stress and promote longevityPLOS ONE

Dear Dr. dos Santos,

Thank you for submitting your manuscript to PLOS ONE. After careful consideration, we feel that it has merit but does not fully meet PLOS ONE’s publication criteria as it currently stands. Therefore, we invite you to submit a revised version of the manuscript that addresses the points raised during the review process.

We look forward to receiving your revised manuscript.

Kind regards,

Manoj Kumar

Academic Editor

PLOS ONE

Reviewers' comments:

Reviewer's Responses to Questions

**Comments to the Author**

1. If the authors have adequately addressed your comments raised in a previous round of review and you feel that this manuscript is now acceptable for publication, you may indicate that here to bypass the “Comments to the Author” section, enter your conflict of interest statement in the “Confidential to Editor” section, and submit your "Accept" recommendation.

Reviewer #1: All comments have been addressed

Reviewer #2: (No Response)

2. Is the manuscript technically sound, and do the data support the conclusions?

Reviewer #1: Partly

Reviewer #2: Yes

3. Has the statistical analysis been performed appropriately and rigorously? 

Reviewer #1: Yes

Reviewer #2: Yes

4. Have the authors made all data underlying the findings in their manuscript fully available?

Reviewer #1: No

Reviewer #2: (No Response)

5. Is the manuscript presented in an intelligible fashion and written in standard English?

Reviewer #1: Yes

Reviewer #2: Yes

6. Review Comments to the Author

Reviewer #1: The work presents new results with scientific relevance. However, several points of the methodology should be reviewed and better explained:

1. The main point to be observed is that different solvents are used in the fruit pulp for its characterization: Methanol:water:formic acid for LC-DAD-MS; acetone, oxalic acid... but for in vivo assays the fruit pulp is solubilized in water.

First question: It is already very well described in the literature that different solvents solubilize very different bioactive compounds. Authors should be careful when attributing results to compounds that may not be present in pulp in water, but which have been identified in pulp in methanol, for example.

Second question: Considering that this is a fruit pulp that has many other food components such as fiber in addition to phenolic compounds, was this pulp really soluble in water? I believe it has not solubilized but formed a suspension. This is confirmed when the authors report that they centrifuge the sample to measure total phenolics and flavonoids. And in in vivo assays, is the suspension or the supernatant used?

I suggest making this issue very clear in the materials and methods. When the pulp diluted in another solvent is used, when the pulp suspended in water is used and when the supernatant is used.

pag 5. Identification for LC-DAD-MS. Which standards and concentrations of curves were used to quantify the compounds? Were the others compounds identified using the library and the mass? make it clear

2. In vivo analyzes using C. elegans.

Some points here are confusing and need to be clarified.

Was the pulp suspended in water or the supernatant used in these experiments?

The mode of exposure of the worms to the pulp should be clearer, sometimes it is described that the worms were exposed to different concentrations of pulp in liquid medium, other times in NGM medium.

Also make it clear when the worms were on food restriction (without bacteria) and for how long. This data is very important and can greatly influence the observed results, as well described in the literature.

pag 10 l. 214-215. Here the authors describe that the worms were exposed to fruit pulp or bacteria in NGM medium until the L4 stage. So the worms exposed to fruit pulp were on food restriction without bacteria?

In the other reproduction, locomotion and lifespan experiments, were the worms in the L4 stage of this treatment used?

Acute toxicity.

The authors report that the acute toxicity was evaluated in liquid medium with worms in stage L4 for 24h and 48h.

Have these L4 stage worms been exposed to fruit pulp before on the plates as described in lines 214-215?

Were the worms in liquid medium for 24h and 48h without bacteria?

The worms in stage L4 stayed in the medium for 24h and 48h and, therefore, they were already in their reproductive period. Therefore, they laid eggs and new worms could be born. Has this been taken into account?

The authors consider the 24h and 48h exposure as acute. Considering that C. elegans has a life cycle of about 20-25 days, can we consider this period as acute? I believe it is better suited as a sub-chronic.

Reproduction, locomotion and lifespan trials.

The authors describe that they used worms in the L4 stage. But were worms already exposed to fruit pulp as described in lines 214-215? Or were they worms exposed in liquid media in the acute test?

During the reproduction and locomotion experiments was the fruit pulp kept on the plate? With or without bacteria as a food source?

Heat stress

The authors report that the fruit pulp was kept on the plates during heating at 37°C. Did these plates also have live bacteria?

Heating was performed in NGM plates or liquid medium.

Make these details clearer in the methodology.

In the discussion, the authors attribute the positive results observed to some compounds present in the fruit pulp. I wonder if the authors really believe that a pulp suspended in water will have the same compounds available for absorption and activity as a fruit pulp dissolved in methanol and acidified with formic acid? Can we attribute these effects to these compounds or others that may not have been quantified or even the presence of fibers and other food compounds in the pulp that also have biological effects?

Reviewer #2: Dears researchers. The manuscript are well performed and written. The changes made in the previous round of review increased the quality the paper.

The only thing that I missed was the discussion of other works involving species from the same genus. For instance, there is research showcasing the antioxidant activity of fruits from the Campomanesia rufa species (a native Brazilian species).

7. PLOS authors have the option to publish the peer review history of their article (what does this mean?). If published, this will include your full peer review and any attached files.

Reviewer #1: No

Reviewer #2: **Yes: **Michele Valquíria Reis

---

## [Author Response · Author response to Decision Letter 1]

12 Sep 2023

Response to reviewers 

 We are very grateful for the comments raised during the review process. All suggestions have been analyzed and the manuscript has been modified. We appreciate the suggestions to improve the final quality of the manuscript. Below we present our statements, and we hope they are sufficient for the requirements of PLOS ONE.

Reviewer #1: The work presents new results with scientific relevance. However, several points of the methodology should be reviewed and better explained:

The main point to be observed is that different solvents are used in the fruit pulp for its characterization: Methanol:water:formic acid for LC-DAD-MS; acetone, oxalic acid... but for in vivo assays the fruit pulp is solubilized in water.

1.1 First question: It is already very well described in the literature that different solvents solubilize very different bioactive compounds. Authors should be careful when attributing results to compounds that may not be present in pulp in water, but which have been identified in pulp in methanol, for example.

Answer: The choice of solvents actually directly impacts the compounds extracted and therefore available to the pulp for effective action. Therefore, all studies carried out in this work were based on lyophilized pulp resuspended in water. In this case, all compounds present in the pulp were present in the resuspended aqueous material. The lyophilized pulp was extremely soluble in water.

The attribution of pharmacological activity to specific compounds, occurred only in the discussion section, lines 580-584 of the revised manuscript (“In this study, FPCA’s protective activity demonstrated against heat stress may have been related to the presence of chemical constituents identified in FPCA, including the flavonoids epicatechin and catechin and their oligomers, procyanidins, which due to their antioxidant and free radical scavenging properties have shown protective effects against heat stress in C. elegans [55].”). In the other cases, the pharmacological activity was attributed generically, that is, to classes of compounds.

The compositions specifically cited were flavonoids, epicatechin and catechin and their oligomers, and procyanidins, which are present in FPCA (lyophilized pulp resuspended in water with excellent solubility). The same compounds are also found using the extraction method with water as solvent (Baldivia et al. 2018, investigated the extract from the stems of S. adstringens using water as a protective solvent, stimulating the presence of flavonoids, epicatechin, catechin, and procyanidins).

The preparation of the sample for LC-DAD-MS analysis was described in detail now since deionized water was first added and maintained in the ultrasonic bath. Subsequently, methanol was added up the rate 7:3 v/v, filtered on Millex® (PTFE membrane, 0.22 µm), and injected. Methanol was added to avoid the precipitation in the chromatographic system. Thus, the composition of the sample analyzed by LC-DAD-MS is similar to the samples evaluated in the biological experiments.

Reference: Baldivia DDS, Leite DF, Castro DTH, Campos JF, Santos UPD, Paredes-Gamero EJ, Carollo CA, Silva DB, de Picoli Souza K, Dos Santos EL. Evaluation of In Vitro Antioxidant and Anticancer Properties of the Aqueous Extract from the Stem Bark of Stryphnodendron adstringens. Int J Mol Sci. 2018 Aug 17;19(8):2432. doi: 10.3390/ijms19082432. PMID: 30126115; PMCID: PMC6121951.

1.2 Second question: Considering that this is a fruit pulp that has many other food components such as fiber in addition to phenolic compounds, was this pulp really soluble in water? I believe it has not solubilized but formed a suspension. This is confirmed when the authors report that they centrifuge the sample to measure total phenolics and flavonoids. And in vivo assays, is the suspension or the supernatant used? I suggest making this issue very clear in the materials and methods. When the pulp diluted in another solvent is used, when the pulp suspended in water is used, and when the supernatant is used.

Answer: From the fruits of Campomanesia adamantium a lyophilized pulp was prepared which obtained excellent solubilization in water (FPCA). This material was resuspended in water to be used in in vitro and in vivo tests. This has been carefully revised in the text. In all assays, including in vivo assays, the FPCA was resuspended in water, using 0.005 g of FPCA in 5 mL of ultrapure water, as described in section Material and Methods, lines 94-95. Only for the measurement of phenolic compounds and flavonoids, the FPCA was centrifuged to use the supernatant as described in lines 115-116 of the revised manuscript as a necessary methodological adaptation.

1.3 pag 5. Identification for LC-DAD-MS. Which standards and concentrations of curves were used to quantify the compounds? Were the others compounds identified using the library and the mass? make it clear

Answer: The identification of the constituents by LC-DAD-MS was performed based on the spectral data (UV, MS, and MS/MS) and they were compared to data published in different manuscripts. Authentic standards were injected to confirm the annotation of some compounds, and they were not applied to quantify them. The quantification evaluations were performed by methods to determine the total phenolic and flavonoid content, lipophilic compounds, and ascorbic acid.

In vivo analyzes using C. elegans.

Some points here are confusing and need to be clarified.

1.4 Was the pulp suspended in water or the supernatant used in these experiments?

Answer: As explained earlier (question 1.2), in all assays, including in vivo assays with the worms C. elegans, FPCA was resuspended in water. For this, 0.005 g of FPCA in 5 mL of ultrapure water was used, using it in its entirety. As described in the Material and Methods section, lines 94-95 of the revised manuscript.

1.5 The mode of exposure of the worms to the pulp should be clearer, sometimes it is described that the worms were exposed to different concentrations of pulp in liquid medium, other times in NGM medium.

Answer: Worms were exposed to FPCA both in solid NGM medium and in liquid M9 medium, in different experiments. This was revised in the text and new information was included in the lines 209-223 of the revised manuscript.

Initially (lines 209-213), it is described that, for the sub-chronic toxicity assay, eggs resistant to alkaline lysis were collected and transferred to Petri dishes containing only NGM culture medium and E. coli (OP50) until reaching the L4 stage. After reaching the L4 stage of development, these worms were transferred to microplates containing M9 liquid medium and subjected to different concentrations of FPCA in the absence of E. coli.

Next (lines 214-223), the tests in which the animals are exposed to FPCA from the egg stage to the L4 stage of development in a solid NGM medium are described. When the worms reach the L4 stage of development in the tests of reproductive and locomotor toxicity, thermal stress, and lifespan, they continue to be maintained in plates containing NGM solid medium, E. coli, and different concentrations of FPCA or water (control). However, for oxidative stress tests, SOD-3, GST-4, and DAF-16 expression, when they reach the L4 phase of development, the worms are transferred to an M9 liquid medium in the absence of E. coli, with different concentrations of FPCA.

1.6 Also make it clear when the worms were on food restriction (without bacteria) and for how long. This data is very important and can greatly influence the observed results, as well described in the literature.

Answer: Information about food restriction has been added on lines 213 and 223 of the revised manuscript. In summary, worms remained without food for 24 and 48 h for the sub-chronic toxicity assay; 6 h for the oxidative stress test; and 30 min for the SOD-3, GST-4, and DAF-16 expression assay.

1.7 pag 10 l. 214-215. Here the authors describe that the worms were exposed to fruit pulp or bacteria in NGM medium until the L4 stage. So the worms exposed to fruit pulp were on food restriction without bacteria?

Answer: We apologize for the text error. The correct information has been inserted on lines 216-218 of the revised manuscript. The worms were kept in a petri dish containing NGM solid medium and E. coli bacteria, being treated with water (control) or different concentrations of FPCA.

1.8 In the other reproduction, locomotion and lifespan experiments, were the worms in the L4 stage of this treatment used?

Answer: For the reproductive and locomotor toxicity tests, and lifespan, the worms were treated from the egg stage of development. After reaching the L4 stage of development, these worms were transferred to new plates containing NGM, E. coli, and FPCA in different concentrations or water (control). This point was revised in lines 218-221 of the revised manuscript, as previously explained (question 1.5). 

Acute toxicity

The authors report that the acute toxicity was evaluated in liquid medium with worms in stage L4 for 24 h and 48 h.

1.9 Have these L4 stage worms been exposed to fruit pulp before on the plates as described in lines 214-215?

Answer: As previously specified (question 1.5), we confirm that for the “sub-chronic toxicity” assay, the worms were exposed to FPCA only after reaching the L4 phase of development, as described in lines 211-213 of the revised manuscript. 

In lines 214-215 indicated, which correspond to lines 216-218 of the revised manuscript, the treatments carried out in other tests are described, namely the tests of reproductive toxicity, locomotor toxicity, response to stress, and longevity.

1.10 Were the worms in liquid medium for 24 h and 48 h without bacteria?

Answer: As previously specified (questions 1.5 and 1.9), for the sub-chronic toxicity test, upon reaching the L4 stage of development, the worms were transferred to microplates containing M9 and different concentrations of FPCA in the absence of E. coli. Thus, the nematodes remained without food for 24 and 48 h. As described in the Material and Methods section, line 228 of the revised manuscript.

1.11 The worms in stage L4 stayed in the medium for 24 h and 48 h and, therefore, they were already in their reproductive period. Therefore, they laid eggs and new worms could be born. Has this been taken into account?

Answer: The posture and hatching of worm eggs during the sub-chronic toxicity test, in the period of 24 and 48 h after the beginning of the treatment did not interfere with the analysis of the data of viable worms. Eggs that do hatch in the M9 medium do not develop beyond the L1 stage due to lack of food. In this way, it is possible to differentiate the evaluated worms from the progeny hatched during the test and this parameter does not interfere with the sub-chronic toxicity test.

1.12 The authors consider the 24 h and 48 h exposure as acute. Considering that C. elegans has a life cycle of about 20-25 days, can we consider this period as acute? I believe it is better suited as a sub-chronic.

Answer: We are grateful for the statement and, although the literature mentions 24 h as acute exposure (Qin et al., 2022), we agree with the observation that 24 and, above all, 48 h would be equivalent to a sub-chronic exposure. Thus, the manuscript was revised and the term “acute toxicity” was changed to “sub-chronic toxicity”, in lines 209, 225, 226, 402, 403, and 408 of the revised manuscript.

Reference: Qin Y, Chen F, Tang Z, Ren H, Wang Q, Shen N, Lin W, Xiao Y, Yuan M, Chen H, Bu T, Li Q, Huang L. Ligusticum chuanxiong Hort as medicinal and edible plant foods: Antioxidant, anti-aging and neuroprotective properties in Caenorhabditis elegans. Front Pharmacol. 2022 Oct 26;13:1049890. doi: 10.3389/fphar.2022.1049890. PMID: 36386171; PMCID: PMC9643709.

Reproduction, locomotion and lifespan trials

1.13 The authors describe that they used worms in the L4 stage. But were worms already exposed to fruit pulp as described in lines 214-215? Or were they worms exposed in liquid media in the acute test?

Answer: As previously specified (question 1.5), we ratify that for the reproductive toxicity, locomotor toxicity, and lifespan tests, the worms were exposed to FPCA from the egg stage to the L4 stage of development in a solid NGM medium. When the worms reach the L4 phase of development in these assays, they continue to be maintained in solid NGM medium with different concentrations of FPCA or water (control). As described in the Material and Methods section, lines 214-223 of the revised manuscript, which corresponded to 214-215.

1.14 During the reproduction and locomotion experiments was the fruit pulp kept on the plate? With or without bacteria as a food source?

Answer: As previously specified (questions 1.5 and 1.13), we ratify that for the reproductive and locomotor toxicity tests, the worms were exposed to FPCA from the egg stage to the L4 stage of development in a solid NGM medium. When the worms reached the L4 phase of development in these assays, they continued to be maintained in plates containing NGM solid medium, E. coli, and different concentrations of FPCA or water (control). As described in the Material and Methods section, lines 214-223 of the revised manuscript, which corresponded to 214-215.

Heat stress

1.15 The authors report that the fruit pulp was kept on the plates during heating at 37°C. Did these plates also have live bacteria? Heating was performed in NGM plates or liquid medium. Make these details clearer in the methodology.

Answer: As previously stated (question 1.5), the worms in the L4 phase of development are kept in plates containing solid NGM medium, E. coli, and different concentrations of FPCA or water (control) which are heated to 37 °C. Although this would be an optimal growth temperature for the bacteria, this is not the case because the bacteria were previously inactivated by kanamycin before being plated onto the NGM medium. Information on the inactivation of bacteria by kanamycin was included in lines 260 of the Materials and Methods section, a revised version of the manuscript. 

1.16 In the discussion, the authors attribute the positive results observed to some compounds present in the fruit pulp. I wonder if the authors really believe that a pulp suspended in water will have the same compounds available for absorption and activity as a fruit pulp dissolved in methanol and acidified with formic acid? Can we attribute these effects to these compounds or others that may not have been quantified or even the presence of fibers and other food compounds in the pulp that also have biological effects?

Answer: The mentioned reagents methanol and formic acid were used only for the chemical analysis by LC-DAD-MS. The pharmacological activities observed in this study result from the FPCA, that is, from the constituents present in the FPCA.

 

Reviewer #2: Dears researchers. The manuscript are well performed and written. The changes made in the previous round of review increased the quality the paper.

The only thing that I missed was the discussion of other works involving species from the same genus. For instance, there is research showcasing the antioxidant activity of fruits from the Campomanesia rufa species (a native Brazilian species).

Answer: We thank you for the observation. The information was included in the discussion section, lines 521-524 of the revised manuscript as described below:

“The presence of phenolic compounds and ascorbic acid has also been verified in another species of the genus Camponamesia, C. rufa, and linked to the antioxidant activity observed by Abreu et al. [32].”

---

## [Editor Report · Decision Letter 2]

15 Oct 2023

PONE-D-23-17391R2*Campomanesia adamantium* O Berg. fruit, native to Brazil, can protect against oxidative stress and promote longevityPLOS ONE

Dear Dr. dos Santos,

Thank you for submitting your manuscript to PLOS ONE. After careful consideration, we feel that it has merit but does not fully meet PLOS ONE’s publication criteria as it currently stands. Therefore, we invite you to submit a revised version of the manuscript that addresses the points raised during the review process.

ACADEMIC EDITOR:I have noted the authors' considerable efforts in revising the manuscript. However, the main body of the manuscript is lacking the inclusion of a flow chart. It is essential to incorporate the flow chart into the main body, preferably within the Materials and Methodology section. Additionally, the conclusion requires further elaboration in accordance with the study's findings, and it should outline potential future directions for the research.

We look forward to receiving your revised manuscript.

Kind regards,

Manoj Kumar

Academic Editor

PLOS ONE

Journal Requirements:

Additional Editor Comments:

I have noted the authors' considerable efforts in revising the manuscript. However, the main body of the manuscript is lacking the inclusion of a flow chart. It is essential to incorporate the flow chart into the main body, preferably within the Materials and Methodology section. Additionally, the conclusion requires further elaboration in accordance with the study's findings, and it should outline potential future directions for the research.

---

## [Author Response · Author response to Decision Letter 2]

18 Oct 2023

Response to academic editor

We are very grateful for the comments raised during the review process. All suggestions made throughout the review process have been analyzed, and the manuscript has been modified. We appreciate the suggestions to improve the manuscript's final quality. Below we present our statements, and we hope they are sufficient for the requirements of PLOS ONE.

ACADEMIC EDITOR:

I have noted the authors' considerable efforts in revising the manuscript. 

1. However, the main body of the manuscript is lacking the inclusion of a flow chart. It is essential to incorporate the flow chart into the main body, preferably within the Materials and Methodology section. 

Answer: The flow chart was previously submitted as a response to the reviewers, but in the current version, it has been included in the main body within the Materials and Methodology section, line 98, of the revised manuscript.

2. Additionally, the conclusion requires further elaboration in accordance with the study's findings, and it should outline potential future directions for the research.

Answer: Potential future research directions have been included in the conclusion of the revised manuscript, lines 622-624.

---

## [Editor Report · Decision Letter 3]

31 Oct 2023

*Campomanesia adamantium* O Berg. fruit, native to Brazil, can protect against oxidative stress and promote longevity

PONE-D-23-17391R3

Dear Dr. Santos,

We’re pleased to inform you that your manuscript has been judged scientifically suitable for publication and will be formally accepted for publication once it meets all outstanding technical requirements.

Kind regards,

Manoj Kumar

Academic Editor

PLOS ONE
---

## [Editor Report · Acceptance letter]

8 Nov 2023

PONE-D-23-17391R3 

*Campomanesia adamantium* O Berg. fruit, native to Brazil, can protect against oxidative stress and promote longevity 

Dear Dr. dos Santos:

I'm pleased to inform you that your manuscript has been deemed suitable for publication in PLOS ONE. Congratulations! Your manuscript is now with our production department. 

Kind regards, 

on behalf of

Dr. Manoj Kumar 

Academic Editor

PLOS ONE